# Genomic innovation of ATD alleviates mistranslation associated with multicellularity in Animalia

Santosh Kumar Kuncha[1,2], Vinitha Lakshmi Venkadasamy[1], Gurumoorthy Amudhan[1], Priyanka Dahate[1], Sankara Rao Kola[1], Sambhavi Pottabathini[1], Shobha P Kruparani[1], P Chandra Shekar[1], Rajan Sankaranarayanan[1]*

[1]CSIR–Centre for Cellular and Molecular Biology, Hyderabad, India; [2]Academy of Scientific and Innovative Research (AcSIR), Ghaziabad, India

**Abstract** The emergence of multicellularity in Animalia is associated with increase in ROS and expansion of tRNA-isodecoders. tRNA expansion leads to misselection resulting in a critical error of L-Ala mischarged onto tRNA[Thr], which is proofread by Animalia-specific-tRNA Deacylase (ATD) in vitro. Here we show that in addition to ATD, threonyl-tRNA synthetase (ThrRS) can clear the error in cellular scenario. This two-tier functional redundancy for translation quality control breaks down during oxidative stress, wherein ThrRS is rendered inactive. Therefore, ATD knockout cells display pronounced sensitivity through increased mistranslation of threonine codons leading to cell death. Strikingly, we identify the emergence of ATD along with the error inducing tRNA species starting from Choanoflagellates thus uncovering an important genomic innovation required for multicellularity that occurred in unicellular ancestors of animals. The study further provides a plausible regulatory mechanism wherein the cellular fate of tRNAs can be switched from protein biosynthesis to non-canonical functions.

*For correspondence: sankar@ccmb.res.in

## Introduction

The ever-growing genome information has shed light on the expansion of the number of tRNA genes, which increases with the complexity of multicellular organisms (*Goodenbour and Pan, 2006*). Such an expansion of tRNA genes led to the appearance of tRNA isodecoders that is tRNAs [with] identical anticodon but with sequence variation(s) elsewhere. This sequence diversification has aided the functional expansion of the tRNA molecules from mere adapters in protein translation to versatile molecules involved in many non-canonical functions. Recent studies have shown the indispensable physiological roles of tRNA and tRNA-derived fragments in diverse functions, such as intergenerational inheritance, RNAi, gene regulation, apoptosis inhibition, ribosome biogenesis, stress granules and also in pathologies such as carcinoma (*Chen et al., 2016*; *Gebetsberger et al., 2017*; *Goodarzi et al., 2015*; *Ivanov et al., 2011*; *Saikia et al., 2014*; *Schorn et al., 2017*; *Sharma et al., 2016*). The tRNA is abundantly saturated with identity elements which are essential for processing, structure, and also performing the canonical functions in translation. However, the emergence of tRNA isodecoders has led to the degeneracy of idiosyncratic features of a few tRNAs, which are essential for the canonical translation function (*Saint-Léger et al., 2016*). Moreover, the advent of multicellularity has led to the production and use of reactive oxygen species (ROS) as signaling molecules, unlike bacteria and yeast that only respond to oxidative stress. It has been suggested that ROS adds the required diversity to the gamut of signaling events involved in cell differentiation and proliferation essential for multicellular systems such HIF pathway and signaling cascade by NRF-2, NFκB, Oct4, p53, etc (*Bigarella et al., 2014*; *Bloomfield and Pears, 2003*;

**eLife digest** The first animals evolved around 750 million years ago from single-celled ancestors that were most similar to modern-day organisms called the Choanoflagellates. As animals evolved they developed more complex body plans consisting of multiple cells organized into larger structures known as tissues and organs. Over time cells also evolved increased levels of molecules called reactive oxygen species, which are involved in many essential cell processes but are toxic at high levels.

Animal cells also contain more types of molecules known as transfer ribonucleic acids, or tRNAs for short, than Choanoflagellate cells and other single-celled organisms. These molecules deliver building blocks known as amino acids to the machinery that produces new proteins. To ensure the proteins are made correctly, it is important that tRNAs deliver specific amino acids to the protein-building machinery in the right order.

Each type of tRNA usually only pairs with a specific type of amino acid, but sometimes the enzymes involved in this process can make mistakes. Therefore, cells contain proofreading enzymes that help remove incorrect amino acids on tRNAs. One such enzyme – called ATD – is only found in animals. Experiments in test tubes reported that ATD removes an amino acid called alanine from tRNAs that are supposed to carry threonine, but its precise role in living cells remained unclear.

To address this question, Kuncha et al. studied proofreading enzymes in human kidney cells. The experiments showed that, in addition to ATD, a second enzyme known as ThrRS was also able to correct alanine substitutions for threonines on tRNAs. However, reactive oxygen species inactivated the proofreading ability of ThrRS, suggesting ATD plays an essential role in correcting errors in cells containing high levels of reactive oxygen species.

These findings suggest that as organisms evolved multiple cells and the levels of tRNA and oxidative stress increased, this led to the appearance of a new proofreading enzyme. Further studies found that ATD originated around 900 million years ago, before Choanoflagellates and animals diverged, indicating these enzymes might have helped to shape the evolution of animals.

The next step following on from this work will be to understand the role of ATD in the cells of organs that are known to have particularly high levels of reactive oxygen species, such as testis and ovaries.

Covarrubias et al., 2008; Lalucque and Silar, 2003; Peuget et al., 2014; Shadel and Horvath, 2015; Sies, 2017).

tRNA synthetases are essential enzymes responsible for maintaining fidelity during protein biosynthesis by selecting both the correct amino acid and tRNA (Ibba and Soll, 2000; Ramakrishnan, 2002). Alanyl-tRNA synthetase (AlaRS) is unique among this class of enzymes in using the universally conserved G3•U70 wobble base pair in the acceptor stem as an identity element for recognition and charging of tRNA^[Ala] (Hou and Schimmel, 1988). Due to the phenomenon of a subtle recognition slippage, the archaeal and eukaryotic AlaRSs have a relaxed specificity for tRNA and can charge alanine on G4•U69 containing (non-cognate) tRNAs also (Sun et al., 2016). Recently, we have shown that G4•U69 is predominant in kingdom Animalia, and specifically in tRNAs coding for Thr (18% in humans and sometimes as high as 43%, in *Takifugu rubripes*), thereby generating L-Ala-tRNA^[Thr](G4•U69) (Kuncha et al., 2018a). Such a tRNA misselection error can result in Thr-to-Ala mistranslation. However, the absence of Thr-to-Ala mutations at the proteome level indicated the presence of a dedicated proofreading factor (Sun et al., 2016). We further showed that L-Ala-tRNA^[Thr](G4•U69) is robustly edited by Animalia-specific tRNA deacylase (ATD) in vitro (Kuncha et al., 2018a).

ATD is a paralog of a 'Chiral Proofreading' enzyme D-aminoacyl-tRNA deacylase (DTD), which is one of the key chiral checkpoints during protein synthesis and present across Bacteria and Eukarya (Kuncha et al., 2019). Earlier studies using ligand-bound crystal structures, in combination with biochemical assays, have shown that DTD operates *via* L-chiral rejection mechanism and as a result also avoids mistranslation of L-Ala to achiral glycine (Ahmad et al., 2013; Kuncha et al., 2018b; Pawar et al., 2017). The L-chiral rejection by DTD is attributed to an active site invariant cross-subunit *Gly-cisPro* motif that acts as a chiral-selectivity filter (Routh et al., 2016). The Gly-Pro (GP) motif

conformation has switched from *Gly-cisPro* in DTD to *Gly-transPro* in ATD. In Animalia, the appearance of tRNA$^{Thr}$(G4•U69) is invariably correlated with the presence of ATD (*Kuncha et al., 2018a*). Thus, ATD is the first and the only known proofreader of tRNA misselection errors. However, the physiological significance and its role in translation quality control are not known. Given the emerging evidence of stress-induced regulation of mistranslation (*Netzer et al., 2009*; *Steiner et al., 2019*), the role of ATD in higher organisms is of immense interest. Here, we show that not only ATD but threonyl-tRNA synthetase (ThrRS) is also involved in correcting the tRNA-induced misselection error by AlaRS. As oxidative stress inhibits the proofreading activity of ThrRS, the protective activity of ATD is indispensable for avoiding mistranslation of Thr codons that is deleterious. Strikingly, the origin of ATD and the G4•U69 containing tRNAs are rooted in Choanoflagellates suggesting its importance for the origin of metazoans at the convergence of multicellularity-induced oxidative stress and tRNA isodecoder expansion.

## Results

### Absence of mistranslation in ATD knockout cells and its ubiquitous expression

Based on the earlier biochemical observations, ATD is expected to avoid Thr-to-Ala mistranslation, which would affect protein homeostasis and also cell survival. Hence to understand the physiological role and relevance of ATD, we generated a CRISPR-Cas9-based ATD knockout of HEK293T cells (*Figure 1A*, *Figure 1—figure supplement 1*). Surprisingly, the ATD null cells did not show any noticeable phenotypic effect which was puzzling, given that ATD is the only known editor of L-Ala-tRNA$^{Thr}$(G4•U69). To further analyze the cells at the molecular levels, Hsp70 was chosen as the marker for proteome stress. Hsp70 is a chaperone which is upregulated in response to protein misfolding and hence used as a marker to study proteostasis stress, and is also implicated in aging and neurodegenerative diseases (*Gupta et al., 2011*; *Hartl, 1996*; *Mosser et al., 2000*; *Rampelt et al., 2012*; *Sala et al., 2017*). The ATD null cells also showed no significant difference in molecular proteome stress marker Hsp70 (*Figure 1B*). Lack of proteome errors is further confirmed by the identification of reporter protein peptides (explained later) using mass spectrometry experiments wherein no missubstitutions of alanine for threonine were observed. The above unexplainable results prompted us to check whether ATD is indeed expressed in vivo. The expression of ATD was initially checked in different cell lines (CHO, Neuro2A, SKOV3, HEK293T, HeLa, NIH/3T3, and mouse embryonic stem cells E14Tg2a), followed by different tissue samples of the mouse. We could see that ATD is ubiquitously expressed in all the cell lines and tissues of mouse tested thus underscoring its physiological importance (*Figure 1C,D*; *Figure 1—figure supplement 2*). The expression data obtained is in line with the available databases such as Human protein Atlas (https://www.proteinatlas.org/), Zfin database (https://zfin.org/), Genecards (https://www.genecards.org/) and Bgee (https://bgee.org/). Despite the ubiquitous presence of ATD, the absence of any noticeable phenotypic and molecular changes in the ATD knockout cell lines prompted us to search for the presence of any other proofreading factor responsible for clearing L-Ala-tRNA$^{Thr}$ in addition to ATD (*Figure 1E*).

### Cross-synthetase error correction in Animalia

The probable proofreading players in the cell that can deacylate L-Ala-tRNA$^{Thr}$(G4•U69) and maintain the fidelity of translation in the absence of ATD include both *cis* (AlaRS, threonyl-tRNA synthetase (ThrRS)) and *trans* editors (AlaXp's and ProX) present in Animalia. Since L-Ala is a cognate amino acid for AlaRS and AlaXp's, the deacylation of L-Ala-tRNA$^{Thr}$ by these editing domains is very unlikely *Beebe et al., 2008*; *Sokabe et al., 2005*). ProX is specific to tRNA$^{Pro}$ and its homolog (Ybak) is known to achieve substrate specificity by forming a binary complex with ProRS, which imparts tRNA specificity for editing (*Ahel et al., 2003*; *An and Musier-Forsyth, 2004*; *An and Musier-Forsyth, 2005*; *Chen et al., 2019*). Thus, the only known and possible player with a high affinity for tRNA$^{Thr}$ is ThrRS. ThrRS has an N-terminal editing domain, which is known to deacylate L-Ser erroneously charged on tRNA$^{Thr}$ (*Dock-Bregeon et al., 2004*). To test whether ThrRS indeed possess deacylation activity towards L-Ala-tRNA$^{Thr}$, we incubated *M. musculus* ThrRS (MmThrRS) with the substrate. Intriguingly, MmThrRS was able to deacylate L-Ala-tRNA$^{Thr}$ at 10 nM concentration but not L-Thr-tRNA$^{Thr}$ (*Figure 2A,B*, *Table 1*). This is the first identified instance where one

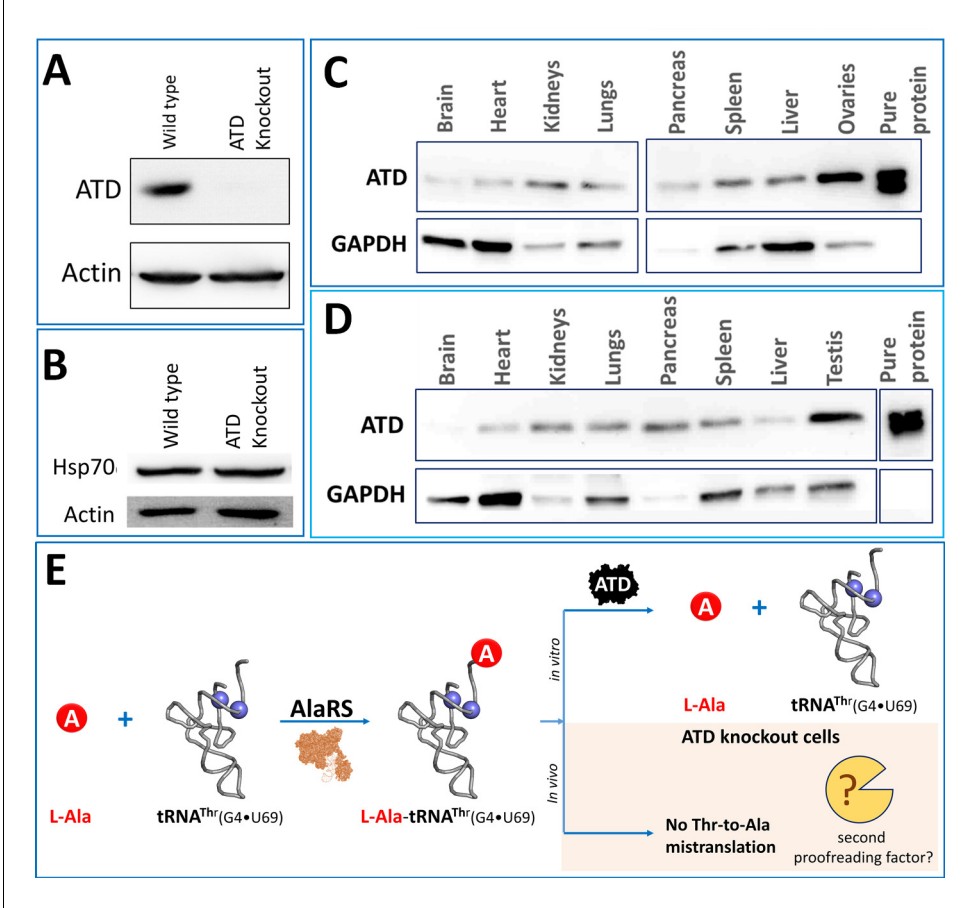

**Figure 1.** Unperturbed proteome homeostasis in *Atd* gene knockout cells and ubiquitous expression of ATD. (**A**) Western blot showing expression of ATD in wild type and ATD knockout cells of HEK293T. (**B**) Western blot of Hsp70 for checking protein misfolding in ATD knockout and wild type of HEK293T cells. No significant change was observed in Hsp70 levels of knockout and wild type. Expression of ATD in different tissues of mice from both (**C**) female and (**D**) male, using GAPDH as a loading control. The tissue samples of mice were lysed in RIPA buffer and quantified using Bradford colorimetric assay and an equal amount of protein was loaded (15 µg per well). (**E**) Schematic representing the tRNA misselection by AlaRS but no mistranslation in the ATD knockout cells indicating the presence of an unknown second proofreading factor for correcting L-Ala-tRNA$^{Thr}$ error.

The online version of this article includes the following figure supplement(s) for figure 1:

**Figure supplement 1.** Generation of ATD knockout.
**Figure supplement 2.** Expression of ATD in different cell lines.

---

aminoacyl-tRNA synthetase, ThrRS in this case, corrects the error caused by another aminoacyl-tRNA synthetase (AlaRS), which misselects the tRNA (*Figure 2C*). These results identified a two-tier functional redundancy in the cell for clearing L-Ala mistakenly attached to tRNA$^{Thr}$ suggesting that ThrRS can correct the tRNA misselection error and hence compensate for the absence of ATD in the knockout cells that do not show any mistranslation of Thr-codons.

## L-Ala-tRNA$^{Thr}$ activity of ThrRS is universally conserved

To validate the universality of the cross-synthetase error correction mechanism, we checked the activity of *D. rerio* (Dr) and *H. sapiens* (Hs) ThrRS on L-Ala-tRNA$^{Thr}$ (*Figure 3A*). In line with the MmThrRS activity, DrThrRS and HsThrRS also deacylated L-Ala-tRNA$^{Thr}$ robustly, indicating that the activity is conserved across higher eukaryotes. Therefore, in Animalia there exists a functional redundancy in clearing L-Ala-tRNA$^{Thr}$ in the form of ATD and ThrRS (*Figure 2C*). It was tempting to check whether the L-Ala activity of ThrRS is conserved across different domains of life or has it been acquired over the course of evolution only in Animalia? It is worth noting here that it is only in

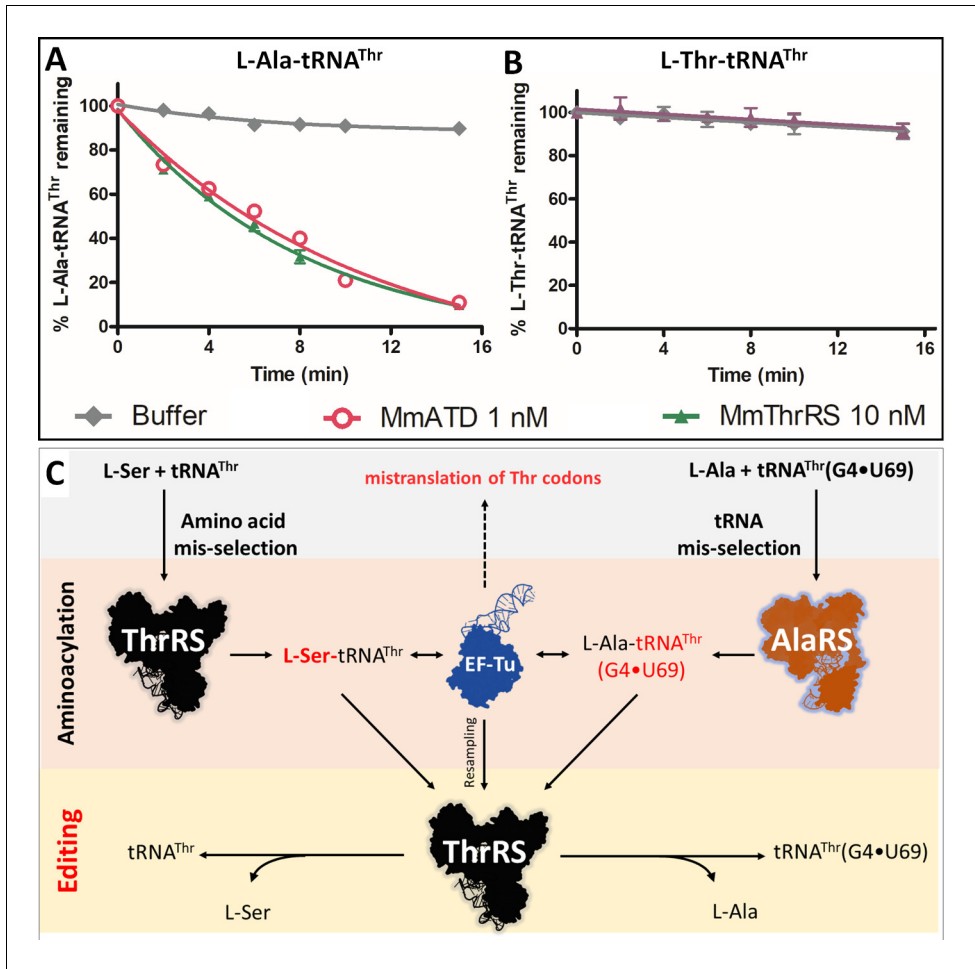

**Figure 2.** Cross-synthetase error correction by ThrRS. (**A**) Deacylation of L-Ala-tRNA$^{Thr}$(G4•U69) by MmThrRS and MmATD, the graph clearly shows that both the enzymes are active on the substrate. (**B**) Deacylation of L-Thr-tRNA$^{Thr}$ by MmThrRS, and as expected MmThrRS has no activity on the cognate substrate. Data are represented as mean ± SD of at least three independent experiments. (**C**) Schematic showing the cross-synthetase error correction by ThrRS. ThrRS mischarges L-Ser on tRNA$^{Thr}$ which is proofread by the editing domain of ThrRS. The tRNA induced misselection by AlaRS generates L-Ala-tRNA$^{Thr}$(G4•U69), which is also proofread by ThrRS editing domain either from the free aminoacyl-tRNA pool or by resampling from EF-Tu.

The online version of this article includes the following source data for figure 2:

**Source data 1.** Deacylation of L-Ala/Thr-tRNA$^{Thr}$ by MmATD and MmThrRS.

Animalia that the error inducing tRNA species is present and hence the problem of L-Ala

**Table 1.** Rate of deacylation by MmATD and MmThrRS.
Deacylation rates of L-Ala-tRNA$^{Thr}$ (200 nM) by MmATD and MmThrRS. The $k_{obs}$ values are calculated by fitting the deacylation curves (*Figures 2A* and *3F*) to the first-order exponential decay equation using GraphPad Prism.

| Enzymes | 5 mM $H_2O_2$ | [Enzyme] | $k_{obs}$ (min$^{-1}$) |
|---|---|---|---|
| MmATD | - | 1 nM | 0.127 ± 0.004 |
| MmATD | + | 1 nM | 0.152 ± 0.005 |
| MmThrRS | - | 10 nM | 0.142 ± 0.004 |
| MmThrRS | + | 10 nM | No activity |

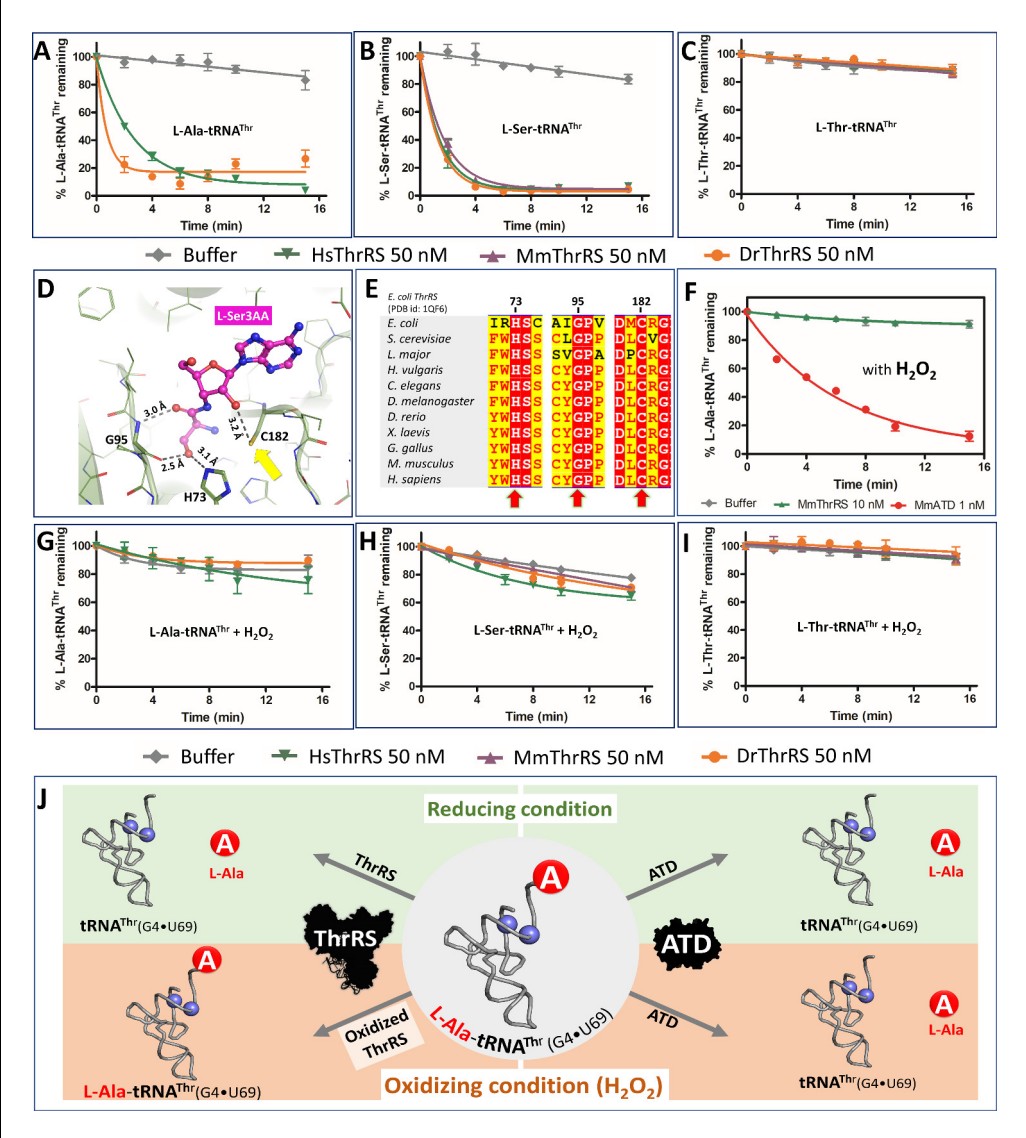

**Figure 3.** ThrRS is sensitive to oxidative stress. (**A–C**) Deacylation of L-Ala/Ser/Thr-tRNA$^{Thr}$ by ThrRS from different organisms in the absence of $H_2O_2$. (**D**) The active site of ThrRS editing domain in complex with L-Ser3AA (magenta sticks) showing the important interaction of the ligand with protein residues (PDB id: 1TKY). The amino acid (carbonyl oxygen and side chain hydroxyl group of L-Ser) of the incoming substrate is captured by an invariant Gly95 and His73. The ROS sensitive Cys182 (indicated using a yellow arrow) interacts with 2'OH of the terminal adenosine A76 of the aa-tRNA. (**E**) Structure-based sequence alignment of the ThrRS editing domain from different organisms, highlighting the invariance of residues His73, Gly95 and Cys182 throughout evolution from bacteria to humans, which are essential for the interaction with the substrate and catalysis. (**F**) Deacylation assay of L-Ala-tRNA$^{Thr}$(G4•U69) by MmThrRS and MmATD in presence of 5 mM $H_2O_2$ in the reaction mixture. The graph clearly indicates that MmThrRS is inactive in the presence of $H_2O_2$ while ATDs activity is unaffected. (**G–I**) Deacylation of L-Ala/Ser/Thr-tRNAThr by ThrRS from different organisms in the presence of 5 mM $H_2O_2$. The biochemical data are represented by the mean ± SD of at least three independent experiments. (**J**) Schematic showing the functional redundancy of ThrRS and ATD in proofreading L-Ala-tRNA$^{Thr}$(G4•U69) in reducing conditions, and inactivation of ThrRS during oxidative stress.

The online version of this article includes the following source data and figure supplement(s) for figure 3:

**Source data 1.** Deacylation of L-Ala/Thr/Ser-tRNA$^{Thr}$ by ATD and ThrRS in presence and absence of oxidative stress.
**Figure supplement 1.** *E. coli* ThrRS proofreads L-Ala-tRNA$^{Thr}$ and is sensitive to oxidative stress.
**Figure supplement 1—source data 1.** ThrRS editing is sensitive to oxidative stress but not ATD.
**Figure supplement 2.** $H_2O_2$ does not affect the synthetase activity of ThrRS.
*Figure 3 continued on next page*

*Figure 3 continued*

**Figure supplement 3.** ATD is inert to oxidative stress.

**Figure supplement 4.** ATD does not contain any cysteine in the active site.

mischarging on tRNA$^{Thr}$. To test this, we checked the deacylation activity of ThrRS from Bacteria. We were surprised to see that bacterial ThrRS from *E. coli* (EcThrRS) was active on L-Ala-tRNA$^{Thr}$, even though neither Bacteria (bacterial AlaRS is discriminatory and charges only tRNAs containing G3•U70 [*Sun et al., 2016*]) nor lower eukaryotes (which lack tRNAs containing G4•U69) possess the problem of tRNA misselection (*Figure 3—figure supplement 1A–C*). Therefore, it is perplexing as to why the L-Ala editing activity of ThrRS is universally conserved. In any case, these biochemical results clearly suggest that the editing site of ThrRS can accommodate and deacylate L-Ala mischarged on tRNA$^{Thr}$, despite lacking the critical interactions emanating from the side chain hydroxyl group of serine (*Figure 3D,E*). Thus, in Animalia, ThrRS maintains the fidelity of decoding Thr codons by performing the dual function of clearing both amino acid (L-Ser-tRNA$^{Thr}$) and tRNA misselection (L-Ala-tRNA$^{Thr}$) errors (*Figure 3A–C*).

## ThrRS editing, but not aminoacylation, function gets inactivated during oxidative stress

Recruitment of a new factor, ATD, even in the presence of ThrRS, a housekeeping gene, suggests the importance of avoiding Thr-to-Ala mistranslation. It raises an important question as to whether this functional redundancy is important for maintaining cellular protein homeostasis during any kind of physiological or stress conditions. Recent thiome studies using HeLa cell lysates show that ThrRS is oxidized at physiological conditions (*Leonard et al., 2009*) and the site of oxidation was found to be an active site invariant cysteine (C182) of ThrRS editing domain, which plays a crucial role in catalysis (*Figure 3D,E*). Oxidation of C182 abolishes the L-Ser-tRNA$^{Thr}$ editing activity of *E. coli* ThrRS has been noted earlier (*Ling and Söll, 2010*). We could establish that oxidizing conditions (using H$_2$O$_2$) do not affect the aminoacylation activity but results in a complete loss of L-Ala-tRNA$^{Thr}$ and L-Ser-tRNA$^{Thr}$ editing activity of MmThrRS (*Figure 3F*; *Figure 3—figure supplement 2*). Interestingly, the synthetase activity of MmThrRS is not affected even in the presence of 10-fold excess (50 mM) of H$_2$O$_2$. To establish the universality of oxidation-induced inactivation of ThrRS editing activity, we tested EcThrRS, DrThrRS, MmThrRS, and HsThrRS (*Figure 3F–I*; *Figure 3—figure supplement 1D–F*). Irrespective of the organism (from bacteria to mammals), in the presence of oxidizing conditions the editing domain of ThrRS is inactive on L-Ala-tRNA$^{Thr}$, however, this did not affect the aminoacylation activity of the enzyme. These deacylation experiments clearly show that the editing site cysteine is prone to oxidation and this feature is conserved across Bacteria and Eukarya.

## ATD is inert towards oxidative stress

Since ThrRS is inactive in the presence of ROS, the obvious question is whether ATD is active in the presence of oxidizing conditions? To this end, we checked for MmATD's activity on L-Ala-tRNA$^{Thr}$ in the presence and absence of H$_2$O$_2$ and found that it was totally unaffected (*Figure 3F*). The inertness of ATD towards oxidative stress was further validated by performing L-Ala-tRNA$^{Thr}$ deacylation assays using ATDs from different organisms such as *Hydra vulgaris* (HvATD) from the invertebrate Animalia, *Danio rerio* (DrATD) from class Pisces, *Gallus gallus* (GgATD) which belongs to Aves and *Homo sapiens* (HsATD) which is a mammal. Irrespective of the origin/class of organisms, all the ATDs were robust in deacylating L-Ala-tRNA$^{Thr}$ even in the presence of oxidizing conditions (H$_2$O$_2$) (*Figure 3J*; *Figure 3—figure supplement 3A,B*). Hence, unlike ThrRS, ATD is insensitive to a significant amount of H$_2$O$_2$. This is in accordance with the absence of any cysteine residue in the active site of ATD, unlike that of ThrRS editing site (*Figure 3—figure supplement 4A,B*).

## ROS induced cellular toxicity in ATD knockout cells

We then asked if ATDs activity is not affected by H$_2$O$_2$, would this factor avoid Thr-to-Ala mistranslation during oxidative stress in vivo? To see the cellular effect and the essentiality of ATD, we initially treated the HEK293T wild type and ATD knockout cells with different concentrations of H$_2$O$_2$ (50, 75 and 100 μM) for 24 hr, followed by cell viability assay (MTT), which revealed that ATD knockout cells

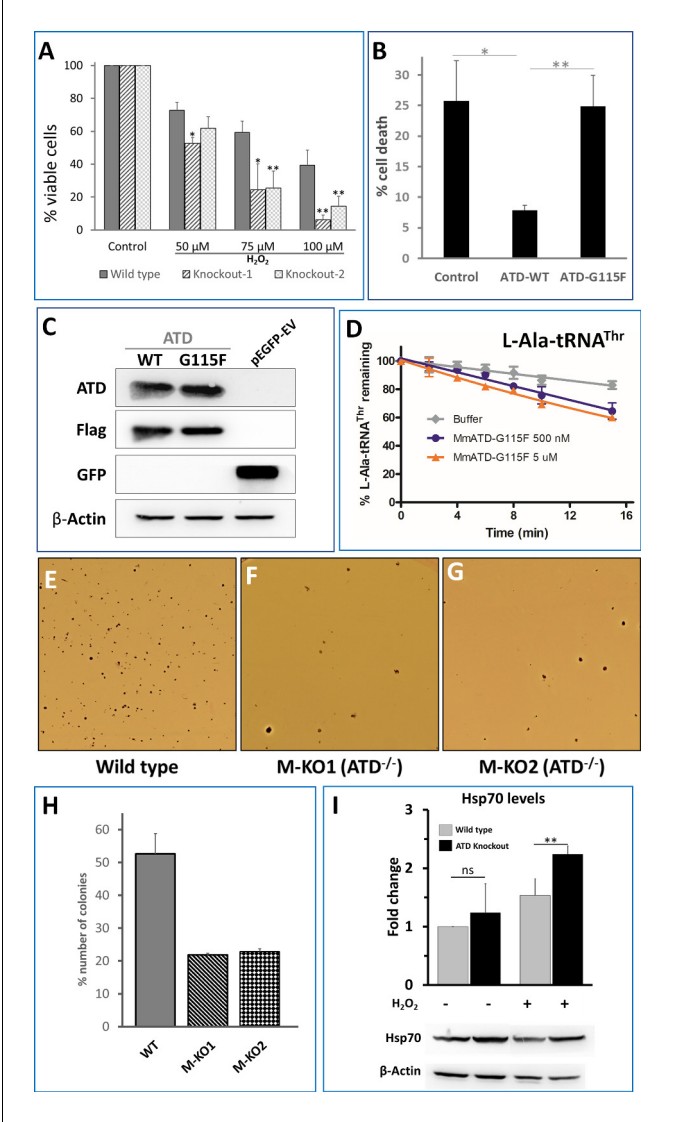

**Figure 4.** ATD is essential during oxidative stress. (**A**) Cell viability assay using MTT assay. Both wild type and knockout cells (knockout 1 and knockout 2 are two lines created using CRISPR-Cas9) are treated with varying concentrations of $H_2O_2$ followed by 24 hr of incubation and assessed for viability. The p-value is a comparison between the wild type and knockout at the same concentration of $H_2O_2$. (**B**) Overexpression of ATD rescues $H_2O_2$ induced cell toxicity: ATD knockout cells of HEK293T were transfected with a vector containing no insert, wild type ATD and inactive mutant of ATD, and treated with 75 µM $H_2O_2$ followed by MTT assay, all the values are normalized using $H_2O_2$-untreated cells as control. * indicates p-value<0.1 and ** indicates p-value<0.01 obtained by doing student's t-test. (**C**) Immunoblotting of ATD knockout cells expressing FLAG-tagged ATD using Anti-ATD and Anti-FLAG antibody. (**D**) Deacylation of L-Ala-tRNA$^{Thr}$ by G115F mutant of MmATD. Data points correspond to the mean of three independent readings and error bars represent one standard deviation. (**E–G**) Leishman-stained colonies of mouse ES cells after growing the wild type and ATD knockout cells in presence of 0.1X βME. (**H**) Graph representing the number of colonies formed by wild type E14Tg2a (mouse embryonic stem cells) and ATD knockout E14Tg2a cells (M-KO represents ATD knockout of MES cells, 1 and 2 refer to the clone number) formed when grown in media containing 0.1X βME compared to that of 1X βME (100 µM). (**I**) Probing for the markers of protein mistranslation, Hsp70, after exposing the wild type and ATD knockout cells to oxidative stress (75 µM of $H_2O_2$) for 6 hr. (p-value for untreated wild type and knockout is insignificant (ns) and the p-value between $H_2O_2$-treated wild type and control is 0.023, denoted by ** obtained by doing student's t-test).

The online version of this article includes the following source data and figure supplement(s) for figure 4:

**Source data 1.** Deacylation of L-Ala-tRNA$^{Thr}$ by G115F mutant of ATD.

**Figure supplement 1.** ATD knockout cells are sensitive to oxidative stress.

*Figure 4 continued on next page*

*Figure 4 continued*

**Figure supplement 2.** G115F mutant of ATD is inactive.
**Figure supplement 3.** ATD knockout of mouse embryonic stem cells.

show pronounced cell death (*Figure 4A*; *Figure 4—figure supplement 1*). We could further show that this is a direct effect of ATD by rescuing the cells from $H_2O_2$ induced toxicity by expressing FLAG-tagged ATD from a plasmid copy (*Figure 4B,C*). To further confirm that the rescue is due to the enzymatic activity of ATD, we used an enzymatically inactive G115F mutant of ATD. This mutant was generated based on our earlier studies on DTD's ligand-bound structures wherein an analogous mutation A112F (PfDTD, PDB id: 4NBI) sterically excludes the binding of adenine (A76) of the incoming substrate (aa-tRNA) and hence renders it inactive (*Ahmad et al., 2013*). As expected, we could show that the G115F mutant of ATD also is similarly inactive for deacylation activity even with 1000-fold excess concentrations (*Figure 4D*; *Figure 4—figure supplement 2*). The G115F mutant of ATD does not rescue ATD knockout cells from the oxidative stress-induced toxicity and therefore unequivocally establishes that it is ATDs enzymatic activity which is essential for the rescue during oxidative stress (*Figure 4B,C*).

To further validate the in vivo results of HEK293T cells, we generated ATD knockout in mouse embryonic stem (mES) cells using the CRISPR-Cas9 paired guide strategy (*Figure 4—figure supplement 3A,B*). mES cells are known to have higher internal ROS and can be cultured only in the presence of antioxidants such as β-mercaptoethanol, which is usually added to the culture media (*Czechanski et al., 2014*; *Han et al., 2008*). We altered the culture conditions by simply decreasing the concentration of the reducing agent, β-mercaptoethanol (βME). mES cells colonies were initially stained using the Leishman stain method and counted under the microscope. Interestingly, in the usual concentration of β-mercaptoethanol (100 μM) the wild type and ATD knockout mES cells were growing normally without any significant change in proliferation and survival. However, at 0.1X (10 μM) concentration of βME, the number of colonies in the knockout cells (from two independently derived clones) was significantly lower compared to that of wild type ((*Figure 4E–H*). While the wild type and knockout have suffered at 0.05X concentration βME and no colonies were observed in the plate (*Figure 4—figure supplement 3C*). These results in combination with HEK293T data unequivocally establish that ATD is essential during physiological conditions of high oxidative stress. It is worth noting here that oxidative stress in ATD$^{-/-}$ cells mimic the scenario of a double knockout condition in which ATD is absent and the editing domain of ThrRS is inactivated. Hence, the hypersensitivity of ATD knockout towards oxidative stress is due to the absence of both the deacylators of L-Ala-tRNA$^{Thr}$ and therefore is expected to result in the mistranslation of Thr codons.

## Perturbation of protein homeostasis in ATD$^{-/-}$ cells under oxidative stress

To check for mistranslation, initially, we set out to look for markers of protein misfolding such as Hsp70. The level of Hsp70 in ATD knockout cells was significantly upregulated (>2 fold) in response to oxidative stress, while the wild type cells showed a marginal upregulation (*Figure 4I*). The mistranslation of Thr codons would also affect the overall cellular proteome homeostasis. To visualize the disturbance in the cellular proteostasis, we used EGFP tagged FlucDM as the sensor of proteome stress (*Gupta et al., 2011*). FlucDM-EGFP aggregates in response to proteome stress and can be visualized either by microscopy or by quantifying its ratio of insoluble to the soluble fraction in the cell. $H_2O_2$-treated ATD knockout cells have more speckles of GFP (due to aggregation of FlucDM) and show a dose-dependent increase in oxidative stress, while the number of aggregates in wild type increases only marginally (*Figure 5A*; *Figure 5—figure supplement 1A*). These observations are further validated by quantifying the ratio of insoluble-to-soluble FlucDM-EGFP using western blots, which showed a marked two-fold increase compared to wild type (*Figure 5B*; *Figure 5—figure supplement 1B*), which is in line with the already observed 1.5 to 2-fold increase in FLucDM insoluble-to-soluble fraction ratio during different stress (heat and proteasome inhibitor) conditions (*Gupta et al., 2011*; *Rawat et al., 2019*). As further evidence to signify the role of ATD in oxidative stress, we observe the upregulation of ATD expression in $H_2O_2$-treated wild-type cells (*Figure 5C*).

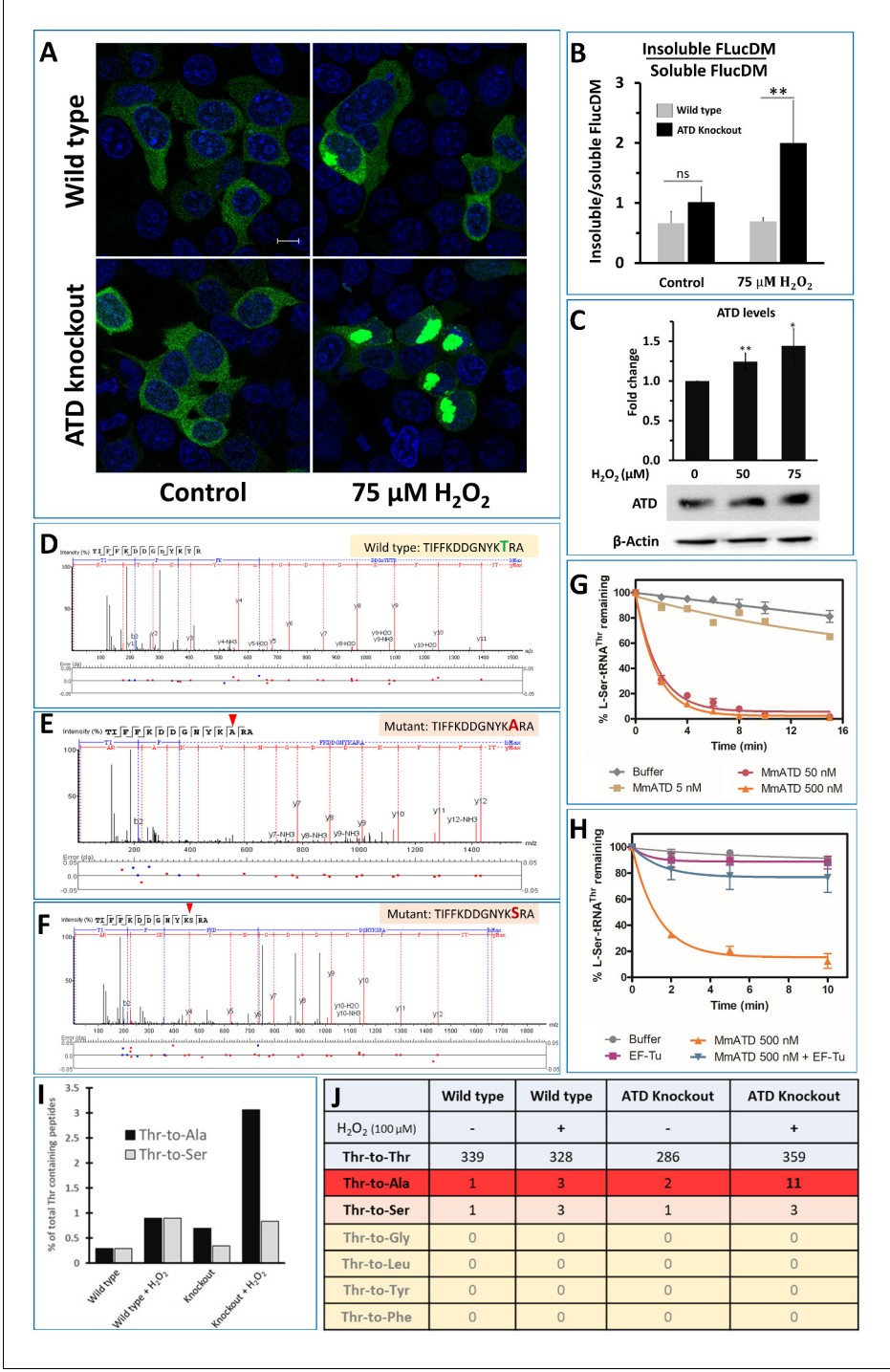

**Figure 5.** ATD avoids ROS-induced mistranslation. (**A**) Microscopic images showing the aggregation of FlucDM-EGFP in the presence and absence of oxidative stress, in wild type and ATD knockout cells. The blue color corresponds to the DAPI staining of the nucleus while green corresponds to the GFP speckles formed due to the aggregation of FlucDM in response to the proteome stress. The scale corresponds to 10 μm. (**B**) Western blot-based quantification of soluble and insoluble (aggregates) fraction of FlucDM-EGFP using anti-GFP antibody. There was no significant difference between untreated wild type and knockout cells, while the treated cells had a significant difference and has a p-value of 0.032 obtained by doing student's t-test indicated by **. (**C**) Overexpression of ATD in response to oxidative stress. Western blot of wild type HEK293T cells lysates after treating the cells with $H_2O_2$ for 2 hr and probed for expression of ATD using specific antibody. There was significant difference in ATD levels between treated and untreated cells, the * indicates a p-value of 0.023 and **

*Figure 5 continued on next page*

*Figure 5 continued*

indicated p-value of 0.016 obtained by doing student's t-test. (**D–F**) Tandem mass spectrometry-based analysis of the reporter protein (overexpressed GFP) isolated from $H_2O_2$-treated ATD knockout cells showing the wild type and mutant (Thr-to-Ser/Ala) spectra of the mistranslated peptides of the reporter protein. (**G**) Deacylation of L-Ser-tRNA$^{Thr}$ by various concentrations of MmATD. (**H**) Deacylation of L-Ser-tRNA$^{Thr}$ by MmATD in the presence and absence of elongation factor Tu. Even 500 nM of MmATD does not deacylate L-Ser-tRNA$^{Thr}$ in presence of elongation factor. This signifies that even though ATD clears L-Ser-tRNA$^{Thr}$ in vitro conditions, in vivo ATD does not deacylate L-Ser-tRNA$^{Thr}$. Biochemical data is mean of at least three independent reading and error bars represent one standard deviation. (**I**) Percentage of peptides with Thr-to-Ala/Ser peptides found compared to that of the wild type peptides. (**J**) Table listing the number of wild type and mutant peptides found in the mass spectrometry experiments and the row highlighted in red corresponds to Thr-to-Ala mutation which is highest in the $H_2O_2$-treated ATD knockout cells.

The online version of this article includes the following source data and figure supplement(s) for figure 5:

**Source data 1.** In vivo ATD edits L-Ala-tRNA$^{Thr}$ but not L-Ser-tRNA$^{Thr}$.
**Figure supplement 1.** Oxidative stress induces proteome stress in ATD knockouts.
**Figure supplement 2.** Oxidative stress leads to Thr-to-Ser mistranslation in wild type cells.

Therefore, ATD is an important factor needed to maintain cellular proteostasis during oxidative stress.

## ATD is essential to avoid Thr-to-Ala substitution in Animalia

The earlier observed proteome stress is very likely due to Thr codons' mistranslation. To pinpoint the exact cause of proteome stress, we overexpressed a reporter protein (GFP using pEGFP vector) with and without oxidative stress, followed by GFP pull-down and subjected to peptide identification using mass spectrometry (MS/MS). The MS/MS data was examined for amino acid substitutions using a GFP-mutant database. We could read increased mistranslation of Thr-to-Ser in a few peptides of samples purified from $H_2O_2$-treated wild type cells (*Figure 5—figure supplement 2*). This quickly rules out the possibility of ATD proofreading L-Ser-tRNA$^{Thr}$, which is in accordance with the low activity of ATD and its likely inability to resample from elongation factor thermo unstable (EF-Tu) (*Figure 5G,H*). However, no toxicity is observed in these cells since the mutation of Thr-to-Ser is a milder substitution and therefore tolerable. In the case of $H_2O_2$-treated ATD knockout cells, the level of Thr-to-Ser mistranslation is similar to that of wild type and further validates our biochemical data that ATD cannot proofread L-Ser-tRNA$^{Thr}$ in vivo. Unlike Thr-to-Ser substitutions, the levels of Thr-to-Ala mistranslation was significantly higher in cells devoid of ATD and treated with $H_2O_2$ (*Figure 5D–F*). As expected, except for Ser and Ala, Thr codons were not mistranslated to other amino acids such as smaller Gly, and bigger Leu, Tyr, and Phe. Since Thr-to-Ala is a drastic change of size, shape, and polarity, it destabilizes the structural integrity of the proteome producing the observed toxicity to the cell. These findings unambiguously provide direct evidence that ATD and ThrRS can proofread L-Ala-tRNA$^{Thr}$ in vivo and the former plays a critical role during oxidative stress.

## Choanoflagellates represent the pre-metazoan ancestry of ATD

ATD is present in kingdom Animalia and so is tRNA$^{Thr}$ possessing G4•U69. Since ATD is a paralog and evolved from a pre-existing DTD, we were curious to find out the first event of emergence or an intermediate of this transition. By extensive bioinformatics search, we have identified that ATD is present in *Salpingoeca rosetta* -a Choanoflagellate (*Figure 6A,B*), but not in other unicellular eukaryotes (like yeast, Plasmodium, Ichthyosporeans, and Filastereans). Choanoflagellates are the colonial unicellular cousins of multicellular animals, and recent studies have shown that these organisms possess genes that are unique to kingdom Animalia and essential for multicellularity i.e., cell cycle regulation (p53), adhesion molecules (cadherin's), hypoxia signaling pathway (prolyl hydroxylase), immune system, sexual reproduction, meiotic division, etc. (*Table 2*; *Brunet et al., 2019*; *Fairclough et al., 2010*; *King et al., 2003*; *Sogabe et al., 2019*; *Woznica et al., 2017*). A thorough phylogenetic analysis of all the available ATD sequences has shown that the emergence of ATD is rooted in Choanoflagellates, which possesses ATD in addition to having a canonical DTD (*Figure 6A*; *Figure 6—figure supplement 1*). Intriguingly, Choanoflagellate ATD has mixed characteristics of both ATD and DTD. The two signature sequence motifs characteristic of all bacterial and

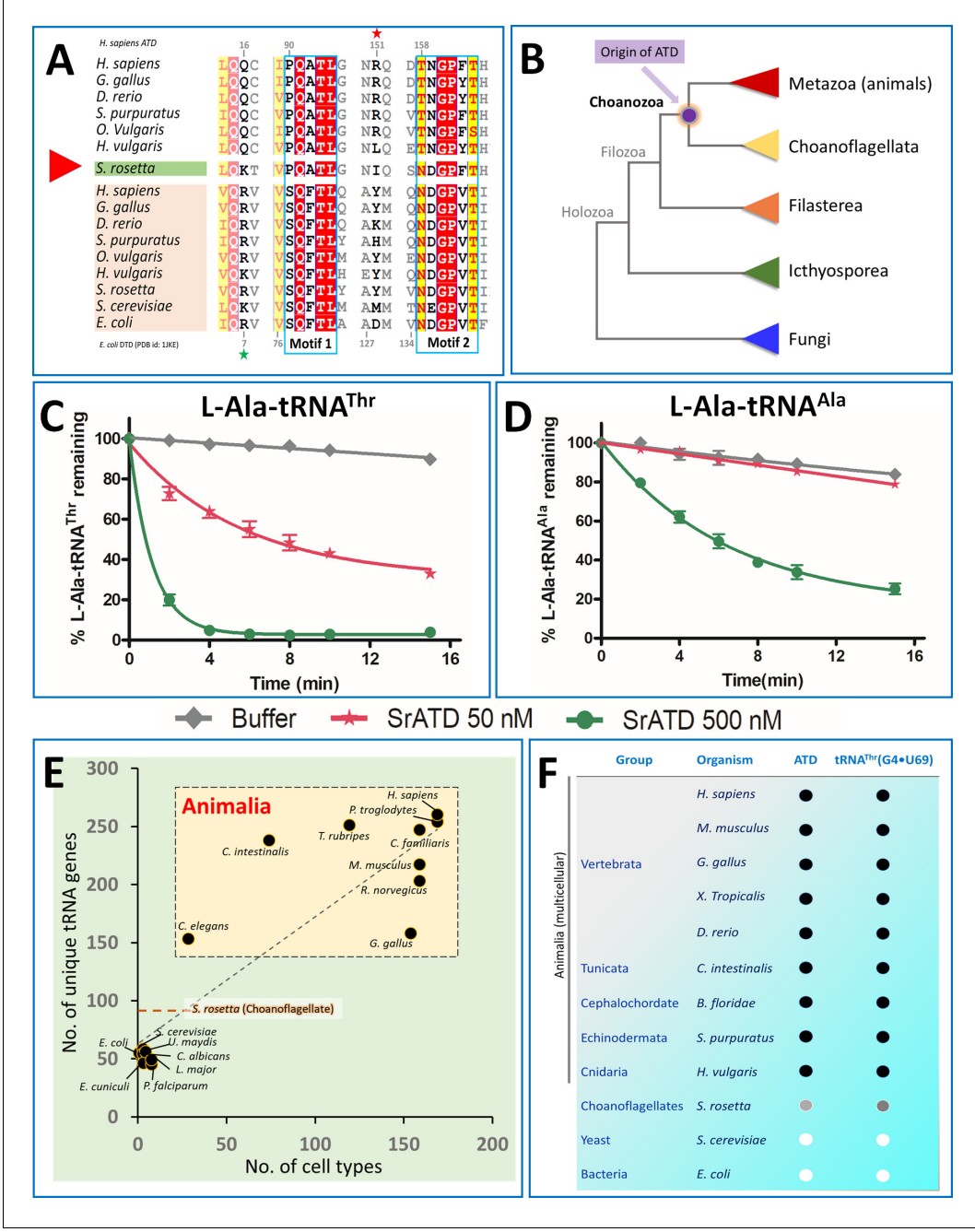

**Figure 6.** Choanoflagellates mark the origin of ATD and G4•U69 containing tRNA isodecoders. (**A**) Structure-based sequence alignment of ATD and DTD sequences across organisms with Choanoflagellate ATD marked in green, which has characteristic motifs from both ATD and DTD. The sequences clearly show the mixed features of Choanoflagellate ATD in having both NXGPXT (DTD-specific) and PQATL (ATD-specific) motif along with a DTD-specific Arg/Lys (Arg7 in *E. coli* DTD, marked with a green star). While lacking the ATD specific Arg151 (marked with a red star). (**B**) The evolutionary tree showing the ATD emergence of ATD in the common ancestors of Choanoflagellates and Animals while absent in other branches of life (except few pathogens such as *Leishmania*, *Trypanosomes, Acanthamoeba,* which might have acquired from the host genome). Deacylation of (**C**) L-Ala-tRNA[Thr] and (**D**) L-Ala-tRNA[Ala] by various concentrations of *S. rosetta* ATD. Biochemical graphs are plotted using mean values and error bars represent the standard deviation. (**E**) Correlation between cell types and number of tRNA genes: Dot plot showing the number of unique tRNA genes and cell types in organisms (taken from *Tan et al., 2009*) across different life forms. The dotted orange line indicates the number of tRNAs in *S. rosetta* obtained using tRNA-Scan (*Chan and Lowe, 2019*) of the genome (*Fairclough et al., 2013*). (**F**) Co-evolution of

*Figure 6 continued on next page*

*Figure 6 continued*

tRNA$^{Thr}$(G4•U69) and ATD: The emergence of ATD is highly linked with the appearance of tRNA isodecoders containing G4•U69. • indicates the presence of the gene, while ○ indicates absence. Choanoflagellates mark the beginning of both ATD and tRNA isodecoders with G4•U69. Intriguingly, the Choanoflagellate ATD is in a transition state and G4•U69 is present in tRNA$^{Gln}$, and not in tRNA$^{Thr}$, and hence indicated with grey circles. The online version of this article includes the following source data and figure supplement(s) for figure 6:

**Source data 1.** Choanoflagellate ATD edits L-Ala-tRNA$^{Thr}$.
**Figure supplement 1.** Choanoflagellates are the root of ATD origin.
**Figure supplement 2.** ATD in transition state.
**Figure supplement 3.** *Hydra vulgaris* tRNAs containing G4•U69.
**Figure supplement 4.** *Salpingoeca rosetta* tRNAs containing G4•U69.
**Figure supplement 5.** Aminoacylation of *S. rosetta* tRNA$^{Gln}$.

eukaryotic DTD are SQFTL and NXGPXT, while PQATL and TNGPY/FTH typify ATD. Choanoflagellate ATD is unusual in having one motif each from ATD and DTD i.e., PQATL and NDGPFT respectively, thus capturing an intermediate in the transition and emergence of a paralog of DTD (*Figure 6A*). Using the available transcriptome and genomics data of different Choanoflagellates, we could identify ATD in a few more Choanoflagellates (*S. macrocollata, S. dolichothecata, S. helianthica, Codosiga hollandica, Mylnosiga fluctuans, and Choanoeca flexa*) (*Brunet et al., 2019*; *Richter et al., 2018*) and all represent the transition state (*Figure 6—figure supplement 2*). Based on the aforementioned mixed features, Choanoflagellate represents a unique case of an intermediate in the metamorphosis of DTD to ATD from the amino acid sequence perspective. We further wanted to check whether this ancestor of Animalia ATD would perform a similar proofreading function. Indeed, biochemical assays showed that *S. rosetta* ATD (SrATD) was active on L-Ala-tRNA$^{Thr}$ thereby proving it to be a functional ATD and not DTD, which is a 'Chiral Proofreader' that does not act on L-aminoacyl-tRNA substrates (*Ahmad et al., 2013*; *Figure 6C*). Interestingly, SrATD was not

**Table 2.** ATD is one of the key genomic innovations that arose in Choanoflagellates which is important for multicellularity.
The symbol • (dot) indicates the presence of the gene, while a white box indicates the absence of the gene in that particular organism. In the case of Porifera and Nematoda which are marked with sky blue color fill, ATD is absent and the tRNA misselection inducing species tRNA$^{Thr}$(G4•U69) is also absent. ATD and tRNA$^{Thr}$(G4•U69) genes are absent in few Annelids and Arthropods hence the boxes are lightly shaded. The data for genes other than ATD are retrieved from *King et al., 2003*; *Nichols et al., 2012*; *Richter et al., 2018*; *Sebé-Pedrós et al., 2010*.

| Process | | Cell cycle regulation | HIF pathway | Immune response | Quality Control |
|---|---|:---:|:---:|:---:|:---:|
| Phylum | Organism | | | | |
| Choanoflagellate | *S. rosetta* | • | • | • | • |
| Porifera | *X. testudinaria* | | | • | • |
| Cnidaria | *H. vulgaris* | • | • | • | • |
| Nematoda | *C. elegans* | • | • | | • |
| Arthropda | *I. scapularis* | • | • | • | • |
| Annelida | *C. teleta* | • | • | • | • |
| Platyhelminthus | *S. mansoni* | • | • | • | • |
| Mollusca | *O. vulgaris* | • | • | • | • |
| Echinodermata | *S. purpuratus* | • | • | • | • |
| Pisces | *D. rerio* | • | • | • | • |
| Amphibia | *X. tropicalis* | • | • | • | • |
| Aves | *G. gallus* | • | • | • | • |
| Reptiles | *N. Naja* | • | • | • | • |
| Mammals | *H. sapiens* | • | • | • | • |
| Protein/Gene | | p53 | PHD | NF-κB | ATD |

only active on L-Ala mischarged on non-cognate tRNA but also could discriminate the correctly charged substrates (L-Ala-tRNA$^{Ala}$) (*Figure 6D*) by 10-fold. The discrimination potential between cognate and non-cognate will be further enhanced in the presence of Elongation factor which binds L-Ala-tRNA$^{Ala}$ strongly compared to that of L-Ala-tRNA$^{Thr}$ (*Kuncha et al., 2018a*; *LaRiviere et al., 2001*). These bioinformatic analysis in combination with biochemical assays provide the basis for the emergence of ATD in the ancestors/cousins of Animalia (*Figure 6B*).

tRNA isodecoder containing G4•U69 also emerged in the unicellular ancestors of animals tRNA misselection by AlaRS is due to the appearance of tRNA isodecoders containing G4•U69 which is in turn linked to the expansion of tRNA genes. The number of unique tRNA genes is highly correlated to the complexity of the organisms (number of cell types) (*Figure 6E*). The presence of G4•U69 in tRNA$^{Thr}$ can be traced from Cnidaria (*Hydra vulgaris*) to the recently evolved mammals (*Homo sapiens*) as is the case for ATD (*Figure 6—figure supplement 3*). Interestingly, Choanoflagellates (*S. rosetta*) represent a stage of transition with 89 unique tRNAs, of which G4•U69 is seen in tRNA$^{Gln}$ and tRNA$^{His}$, but not in tRNA$^{Thr}$ (*Figure 6—figure supplement 4*). Since in eukaryotes, tRNA$^{His}$ undergoes a post-transcriptional modification of adding guanine at the −1 position, which acts as a major determinant for histidyl-tRNA synthetase and a negative determinant for rest of the synthetases, and therefore would make it inert towards AlaRS (*Giegé et al., 1998*; *Jackman and Phizicky, 2006*; *Tian et al., 2015*). However, the presence of G4•U69 in tRNA$^{Gln}$ possibly necessitates that Choanoflagellates have a proofreading factor that can edit L-Ala erroneously charged on tRNA$^{Gln}$(G4•U69). We could show that tRNA$^{Gln}$(G4•U69) could be charged by AlaRS albeit at much lower levels of 4% to 7% than observed for tRNA$^{Thr}$(G4•U69) at 30% to 40% (*Figure 6—figure supplement 5*). Due to the low levels of aminoacylation of Choanoflagellate tRNA$^{Gln}$(G4•U69), we could not perform the deacylation assays using L-Ala-tRNA$^{Gln}$. In any case, even such low levels of tRNA misselection would be precarious and Choanoflagellate ATD, which is in transition, is expected to act on these substrates and this aspect requires further probing. Therefore, the presence of ATD strictly alongside tRNA isodecoders containing G4•U69 even in Choanoflagellates, but not in fungi or other protists, underscores that ATD's emergence is strongly linked with the appearance of tRNA isodecoders containing G4•U69 (*Figure 6F*).

## Discussion

ROS is an inevitable cellular metabolite of increased metabolism in animals, produced by mitohormesis, NOX enzymes and also β-oxidation of lipids in peroxisomes (*Balaban et al., 2005*; *Lambeth, 2004*; *Mohanty and McBride, 2013*). Unlike bacteria and lower eukaryotes, in Animalia ROS is an important signaling molecule and implicated in many cellular pathways (activating NFκB, NRF2, p53, Oct4, etc.) and also for the origin of multicellularity (*Bigarella et al., 2014*; *Bloomfield and Pears, 2003*; *Covarrubias et al., 2008*; *Lalucque and Silar, 2003*; *Peuget et al., 2014*; *Shadel and Horvath, 2015*; *Sies, 2017*). Cells avoid or nullify the toxic effects of ROS on protein quality control either by improving the fidelity of aminoacyl-tRNA synthetase or by increasing the overall Met content in the cellular proteome (*Goodenbour and Pan, 2006*; *Ling and Söll, 2010*). In the case of proofreading tRNA misselection, the functional redundancy of editing L-Ala-tRNA$^{Thr}$ by ThrRS and ATD is broken in the presence of oxidative stress. As mentioned earlier, thiome studies have shown that Cys182 (residue number corresponds to *E. coli* ThrRS), which is essential for editing activity, gets oxidized at physiological conditions (cellular ROS). Since oxidative stress is an important component of multicellular systems, the innovation of ATD in Choanoflagellates seems to be essential to avoid the deleterious effects of tRNA misselection on the cellular proteome. Choanoflagellates mark the origin of many Animalia-specific genes which are essential for multicellularity (*Brunet and King, 2017*; *Brunet et al., 2019*; *Fairclough et al., 2010*; *Richter et al., 2018*; *Sogabe et al., 2019*; *Young et al., 2011*). Therefore, the emergence of this metazoan specific proofreader, ATD, in Choanoflagellates is strictly linked to the appearance of tRNA isodecoders containing G4•U69 and underscoring a strong coevolution of these two, thus implicating their role in the emergence of multicellularity (*Figure 6E,F* and *Figure 7*).

Recent studies have demonstrated the versatile roles of tRNA and tRNA-derived fragments in multiple cellular functions such as regulation, sperm fertility, intergeneration inheritance, and integrated stress response (*Doowon et al., 2018*; *Fricker et al., 2019*; *Gebetsberger et al., 2017*; *Nätt et al., 2019*; *Schwenzer et al., 2019*). The appearance of tRNA$^{Thr}$ isodecoders with G4•U69 in

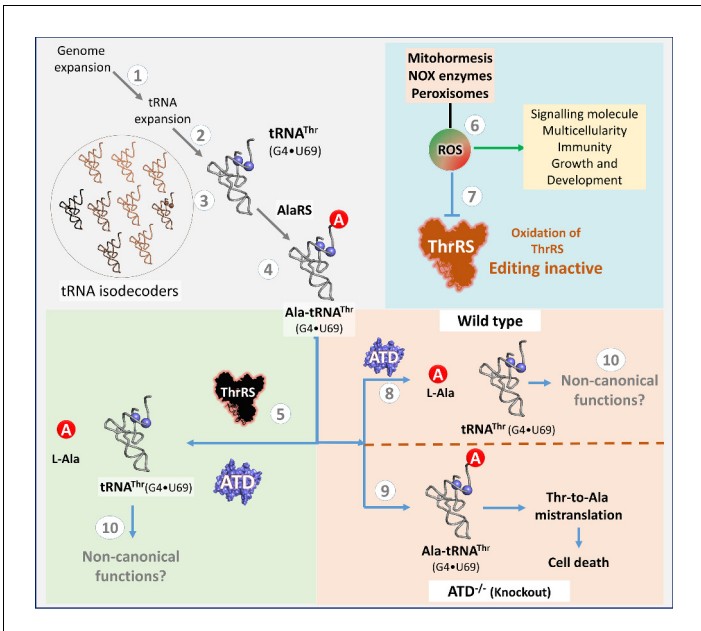

**Figure 7.** The confluence of tRNA isodecoder expansion and oxidative stress necessitates the recruitment of ATD in kingdom Animalia. (1) Genome expansion leads to the appearance of (2,3) tRNA isodecoders which are used in different canonical and non-canonical functions, of which tRNA$^{Thr}$(G4•U69) is (4) mischarged by AlaRS, to give the tRNA misselection product L-Ala-tRNA$^{Thr}$(G4•U69). (5) in vitro these misselection products are cleared by ThrRS and ATD. 6) In multicellular animals, ROS acts as an important cellular metabolite performing various physiological functions by triggering the appropriate signaling pathway. 7) ROS generated internally in cells oxidizes the active site cysteine of ThrRS editing domain thereby affecting its proofreading activity on L-Ala-tRNA$^{Thr}$. 8) To avoid mistranslation ATD is recruited in Animals and Choanoflagellates as it is inert towards ROS and robustly proofreads L-Ala-tRNA$^{Thr}$ to avoid (9) Thr-to-Ala mistranslation at the proteome level followed by cell death. 10) The tRNA isodecoders that are deacylated by ATD can either go for fresh aminoacylation or can be rerouted for other non-canonical functions.

Animalia, and its conservation at the cost of tRNA misselection suggests a strong physiological role. The presence of 3' tRNA-derived fragments (tRF id: 3020a and 3020b) of the tRNA$^{Thr}$(G4•U69) in the tRF database (http://genome.bioch.virginia.edu/trfdb/) (*Kumar et al., 2015*) indicates the possible role of these isodecoders in non-canonical functions. One of the unknown puzzles in RNA biology is how these tRNAs are effectively routed for non-canonical functions. Post-transcriptional modifications of tRNA are implicated in regulating the tRNA folding and also fragments generation (*Durdevic and Schaefer, 2013*; *Lyons et al., 2018*; *Pan, 2018*). However, tRNAs once aminoacylated are channeled to the ribosome by elongation factor. Therefore, the presence of deacylators like ATD and ThrRS can strip the non-cognate amino acid from these tRNAs and thus provides one of the plausible ways of diverting a part of the tRNA pool away from translation (*Figure 7*). It appears that the emergence of ATD as a translational quality control factor in Choanoflagellates has been utilized to selectively regulate the free tRNA pool for other non-canonical functions in Animalia, which remains to be explored.

Similar to genetic variations allowing to adapt/provide an advantage in stress conditions, the generation of variations at the proteome levels are known to be advantageous (*Kelly et al., 2019*; *Mohler and Ibba, 2017*). In lower systems such as *Mycoplasma*, mistranslation has been shown to be advantageous, wherein the presence of editing defective aaRSs (LeuRS, ThrRS, PheRS) allows the organism to gain phenotypic plasticity by generating proteome diversity (*Li et al., 2011*; *Mohler and Ibba, 2017*; *Pezo et al., 2004*). The current work involves the use of acute oxidative stress (H$_2$O$_2$) and further prompts to look at the behavior of ATD knockout cells in the presence of chronic oxidative stress such as activators of NADPH oxidase. ROS-induced inactivation of the editing activity of ThrRS can be used as a conserved mechanism for generating subtle variations in the cellular proteome. At the same time, the spatiotemporal regulation of ATD in combination with

oxidative stress can generate a huge repertoire of proteome diversity that can be beneficial under certain cellular scenarios. ATD's expression levels are very likely modulated by the level of oxidative stress as seen in the case of reproductive tissues (*Figure 1C,D*), which also suggests a mode of regulation via a feedback mechanism. It is worth noting here that the appearance of multicellular animals ~ 750 million years ago is also marked by a significant increase in the global atmospheric oxygen levels which moved from <10% to nearly 20% (*He et al., 2019*; *Kump, 2008*). Overall, the confluence of tRNA expansion (*Figure 7*) and oxidative stress, both external and internal, necessitates the emergence of ATD to maintain cellular proteostasis in Animalia.

# Materials and methods

## Key resources table

| Reagent type (species) or resource | Designation | Source or reference | Identifiers | Additional information |
|---|---|---|---|---|
| Strain, strain background (*Escherichia coli*) | BL21(DE3) | GE Healthcare | GE27-1542-01 | - |
| Strain, strain background (*Escherichia coli*) | Rosetta(DE3) | Sigma-Aldrich | 70954 | - |
| Cell line (*Homo sapiens*) | HEK293T | ATCC | CRL-3216 (RRID:CVCL_0063) | - |
| Cell line (*Mus musculus*) | E14Tg2a | ATCC | CRL-1821 (RRID:CVCL_9108) | - |
| Antibody | Anti-ATD (Rabbit and polyclonal) | Invitrogen | PA524053 (RRID:AB_2541553) | 1:2000 |
| Antibody | Anti HSP-70 (Mouse and monoclonal) | Abcam | ab47454 (RRID:AB_881521) | 1:5000 |
| Antibody | Anti-GFP (Rabbit and monoclonal) | CST | 2956S (RRID:AB_1196615) | 1:5000 |
| Antibody | Anti-GAPDH (mouse and Monoclonal) | Abcam | ab8245 (RRID:AB_2107448) | 1:2000 |
| Antibody | Anti-β-actin (Rabbit and monoclonal) | Santacruz | Sc47778 (RRID:AB_2714189) | 1:2000 |
| Antibody | Anti-mouse IgG (Rabbit and monoclonal) | GE Healthcare | NA931-1ML (RRID:AB_772210) | 1:2000 |
| Antibody | Anti-rabbit IgG (Mouse and monoclonal) | Santacruz | sc-2357 (RRID:AB_628497) | 1:10000 |
| Recombinant DNA reagent | *Escherichia coli* ThrRS (Plasmid) | PMID:26113036 | UniProt id: P0A8M3 | Cloned in pET28b (N-terminal 6X-His tag) |
| Recombinant DNA reagent | *Danio rerio* ThrRS (Plasmid) | This paper | Uniprot id: A2BIM7 | Cloned in pET28b (N-terminal 6X-His tag) |
| Recombinant DNA reagent | *Homo sapiens* ThrRS (Plasmid) | This paper | Uniprot id: P26639 | Cloned in pET28b (N-terminal 6X-His tag) |

*Continued on next page*

*Continued*

| Reagent type (species) or resource | Designation | Source or reference | Identifiers | Additional information |
|---|---|---|---|---|
| Recombinant DNA reagent | *Mus musculus* ThrRS (Plasmid) | PMID:29579307 | - | Cloned in pET28a (N-terminal 6X-His tag) |
| Recombinant DNA reagent | *Hydra vulgaris* ATD (Plasmid) | This paper | - | Cloned in pET28b (C-terminal 6X-His tag) |
| Recombinant DNA reagent | *Salpingoeca rosetta* ATD (Plasmid) | This paper | - | Cloned in pET28b (C-terminal 6X-His tag) |
| Recombinant DNA reagent | FlucDM-EGFP (Plasmid) | PMID:21892152 | - | - |
| Recombinant DNA reagent | *Mus musculus* ATD (Plasmid) | This paper | Uniprot id: Q8BHA3 | Cloned in pEGFP-N1 (C-terminal FLAG tag) |
| Recombinant DNA reagent | *Mus musculus* ATD G115F (Plasmid) | This paper | - | Cloned in pEGFP-N1 (C-terminal FLAG tag) |
| Commercial assay or kit | Lipofectamine 3000 | ThermoFisher Scientific | L3000015 | - |

## Biochemistry

All the components for biochemical assays were generated as mentioned in *Kuncha et al., 2018a*. The sequence coding for threonyl-tRNA synthetase (*Homo sapiens, Danio rerio,* and *Escherichia coli*) was amplified from cDNA for human and zebrafish and from genome for *E. coli* and cloned into pET28b with an N-terminal 6X His-tag. All the proteins were expressed in *E. coli* BL21-CodonPlus (DE3)-RIL strain, except *E. coli* ThrRS, which was expressed in *E. coli* BL21. All the proteins were purified using the affinity-based column (Ni-NTA), followed by size exclusion chromatography. Purified proteins were stored at $-30°C$ in 150 mM NaCl, 200 mM Tris (pH 7.5) and 50% glycerol. Genes coding *M. musculus* tRNA$^{Thr}$(G4•U69), tRNA$^{Ala}$ is in vitro transcribed using MEGAshortscript T7 Transcription Kit (Thermo Fisher Scientific, USA), followed by 3' end labeling using CCA-adding enzyme (*Ledoux and Uhlenbeck, 2008*). Alanylation of tRNA$^{Thr}$(G4•U69) was done using *M. musculus* alanyl-tRNA synthetase. EF-Tu activation experiments were performed using pyruvate kinase and phosphoenolpyruvate as explained in *Routh et al., 2016*. Deacylations were performed using a range of enzyme concentrations, while the concentration which gave a gradual curve was used for curve fitting in GraphPad Prism software, and each data point in the graph represents mean of at least three reading, while the error bars represent the standard deviation from the mean.

## Cell culture, transfection, and microscopy

HEK293T cells were acquired from ATCC and confirmed using STR profiling. HEK293T cells were cultured and maintained in DMEM containing 10% fetal bovine serum, 50 IU ml$^{-1}$ penicillin, and 50 µg ml$^{-1}$ streptomycin. Cultures were grown in a humidified incubator at 37°C and 5% CO$_2$. Transfections were done using Lipofectamine 3000 reagent (Invitrogen) according to the manufacturer's manual. A routine check for mycoplasma contamination was checked using DAPI staining. For microscopy the cells were grown on coverslips, fixed with 10% formaldehyde, permeabilized with Triton X100, and stained with DAPI (Sigma). The cells were imaged in Zeiss Axioimager Z.1 Microscope or Leica TCS SP8.

Mouse ES cells (E14Tg2a) were grown as described in *Jana et al., 2019*. Briefly, cells were cultured on tissue culture-treated plates/dishes coated with 0.1% gelatine, in GMEM media supplemented with L- glutamine, 100 µM β-mercaptoethanol, 1 mM non-essential amino acids (Gibco), 100

units/ml human LIF supplemented with 10% fetal bovine serum (Gibco). Cultures were grown in a humidified incubator at 37°C and 5% $CO_2$. mES cells were transfected using P3 Primary cell 4D-Nucleofector Kit. For colony formation assay of ES cells, 200 cells per well (6 well plate) were plated and Leishman staining was done after 5 days of growth, the cell were grown in media containing a variable concentration of β-mercaptoethanol.

## Knockout generation

For generating knockout in HEK293T, two SgRNAs were designed to target the intron1 and exon1 region of the *Atd* gene, the sequences of the gRNA are sgRNA1-5' TTCGTCGTGCCCCGCCTCGTC 3' and SgRNA2-5' CAGATCGCGTCGAATTCCCC 3'. The SgRNA oligos were cloned into the BbsI sites of pU6-(BbsI)-CBh-Cas9-T2A-iRFP670 and verified by sequencing. The U6-SgRNA1-gRNA scaffold cassette was amplified and cloned into the XbaI site of the pU6-(BbsI)-CBh-Cas9-T2A-iRFP670 plasmid contains the sgRNA2 to generate a dual sgRNA plasmid. These dual SgRNA plasmids were transfected into HEK293T cells, followed by cell sorting to enrich for transfected cells expressing iRFP670. The sorted cells were cultured and clones were established by dilution cloning. The genomic DNA from replica plates of the clones were genotyped by PCR using primers flanking the sgRNAs regions to detect the deletion in ATD locus. The forward primer 5' CACTGAGCGCCTTC TACAGAGTTG 3' and reverse primer 5' GAAAGTAGAAGGAACTCATAGTGAC 3'. The ATD knockouts were further validated by sequencing the PCR amplicon and by performing western blot for ATD protein.

Sg RNA oligos for mouse ES cell knockouts were designed at the exon1 and intron1 junction of the gene coding for ATD, the sequences of the oligos are 5' GGCCGATGGAGACGCCGCGG 3' and 5' CCCTATCCGCGGAACCGTGC 3'. The protocol for knockout generation in ES cells is identical to HEK293T cells, except that the plasmid was transfected into the cells using nucleofection. The mES cell colonies were screened using gene-specific primers 5' CAAAGCTGGTCAATTCCACATCCG 3' and 5' ATTCTGAGAAGCGAGATGGCTCAC 3' which flank the target region.

## MTT assay

The wild type and ATD knockout HEK293T cells were plated in 12 well culture plates (50000 cells/well) and incubated for 24 hr in $CO_2$ incubator. The following day different concentrations of $H_2O_2$ (50, 75, and 100 μM) were added to the corresponding wells and incubated for 24 hr. After the incubation time, 0.5 mg/ml of MTT solution was added to each well and incubated in dark for 3 to 4 hr. The Formazan crystals were dissolved by adding 500 μl of 10% acidified-SDS, followed by transfer to 96 well plates. The intensity of the color was measured using a spectrophotometer at 562 nm. As a confirmatory step the number of live cells per well were counted under 10X objective, and 10X eyepiece magnification of compound microscope using Neubauer-improved counting chamber (Paul Marienfeld GmbH and Co. KG, Germany), cells were stained using trypan-blue.

## Western blotting

In general, wild type and knockout cells (treated and untreated) are harvested, washed with PBS for 2 times and lysed using Laemmli buffer, boiled and separated on SDS-PAGE and transferred to PVDF membrane, followed by probing with appropriate primary and secondary antibody. In the case of luciferase experiments, wild type and knockout cells were transfected with FlucDM-EGFP constructs and treated with different concentrations of $H_2O_2$. Following the treatment, these cells were lysed using NP-40 containing lysis buffer [50 mM Tris–HCl (pH 7.8), 150 mM NaCl, 1% NP-40, 0.25% sodium deoxycholate, 1 mM EDTA, protease inhibitor cocktail (Roche)] and centrifuged at 12,000 g at 4°C for 15 min, the supernatant was used as the soluble fraction and the pellet was boiled in Laemmli buffer that was used as the insoluble fraction and was subjected to immunoblotting.

## Sample preparation for mass spectrometry

For purifying the reporter protein, GFP, the cells were transfected with pEGFP-N1 empty vector and allowed to grow for 24 hr, followed $H_2O_2$ treatment. Cells were initially washed with PBS and followed by lysis using a buffer containing 10 mM Tris (pH 7.8), 150 mM NaCl, 0.5 mM EDTA, 2 mM $Na_3VO_4$, 10 mM NaF, 10 mM N-ethylmaleimide, protease inhibitor mixture, and 0.5% Nonidet P-40.

The lysates were subjected to centrifugation at 20,000 g, 10 min at 4°C, the supernatant was incubated with GFP-Trap beads (ChromoTek) for 2 hr at 4°C. The beads were washed with buffer containing ingredients of lysis buffer except for Nonidet P-40, to remove non-specifically bound proteins. To the GFP bound beads Laemmli sample buffer is added, boiled, and loaded on the SDS-PAGE. After running the PAGE gel followed by coomassie staining, regions containing the protein are sliced and subjected to reduction, alkylation, and in-gel digestion using Trypsin as described by *Shevchenko et al., 2006*. Finally, the peptides were desalted and enriched as per the protocol in *Rappsilber et al., 2007*.

## Mass spectrometry and analysis

The Q Exactive HF (Thermo Fisher Scientific, Bremen, Germany) was used to perform HCD mode fragmentation and LC-MS/MS analysis. Samples were fractionated using commercial column Pep-Map RSLC C18, 3 μm, 100 Å, 75 μm i.d. ×150 mm. The LCs used were EASY-nLC 1200 systems (Thermo Fisher Scientific, San Jose, CA). The column temperature was maintained at 30°C. The peptides were loaded in solvent A (5% Acetonitrile 0.1% formic acid in Ultrapure water) and eluted with a nonlinear gradient of solvent B (0.1% TFA, 0.1% formic acid in 95% acetonitrile) by a gradient to 25% of solvent B over 35 min and 40% for 10 mins at a constant flow rate of 300 nL/min. Instruments were configured for DDA using the full MS/DD−MS/MS setup. Full MS resolutions were set to 60000 at m/z 200, and full MS AGC target was 3E6 with an IT of 100 ms. The mass range was set to 400–1650. AGC target value for fragment spectra was set at 1E5, and the intensity threshold was kept at 3E4. Isolation width was set at 1.3 m/z. The normalized collision energy was set at 28%. Peptide match was set to preferred, and isotope exclusion was on.

The data were analyzed in two software's PEAKS6.0. and Proteome Discoverer 2.2. Peaks 6.0 De novo sequencing and database search analysis parameters: Trypsin, and 3 allowed missed cleavages in one peptide end. The parent mass tolerance of 10 ppm using monoisotopic mass, and fragment ion tolerance of 0.05 Da. Carbamidomethylation of cysteine (+57.02) as a fixed modification, methionine oxidation (+15.99) and deamidation of asparagine and glutamine (NQ +0.98) were set as variable modifications. Data was validated using a false discovery rate (FDR) method. Peptide identifications were accepted for peptides with −10logP score cut-off was set to >20 to achieve theoretical peptide FDR of 0.1%.

Proteome Discoverer is an MS data analysis platform provided by Thermo Fisher Scientific for its mass spectrometers. Raw files of GFP control and knockout samples were imported into Proteome Discoverer 2.2, and HTSequest algorithm search was used. In both the software, the sample was searched against the in-house mutant database. The enzyme specificity was set to trypsin, and three missed cleavages were allowed. Carbamidomethylation of cysteine was set as fixed modification and oxidation of methionine as a variable modification. The identified peptide sequence with more than threshold scores (−10logP score >50 for peaks 6.0) were used to BLAST against wild type GFP protein sequence to find out mutations.

## Sequence analysis and image generation

All the sequences are extracted using the Protein-BLAST search and the structure-based sequence alignments are generated using the T-Coffee server (*Notredame et al., 2000*). Phylogenetic trees were constructed using iTOL server (https://itol.embl.de/). Images of different protein structures are generated using PyMOL. While the tRNA sequences are taken from GtRNAdb (*Chan and Lowe, 2016*).

## Acknowledgements

The authors thank Swasti Raychaudhuri (CSIR-CCMB, India) for providing the FlucDM-EGFP construct, Xiao-Long Zhou (University of Chinese Academy of Sciences, Shanghai, China) for providing MmThrRS construct. The authors thank Richa Khanna (CSIR-CCMB, India) and Hanuman T Kale (CSIR-CCMB, India) for help with cell culture experiments.

# Additional information

## Competing interests

Rajan Sankaranarayanan: Reviewing editor, *eLife*. The other authors declare that no competing interests exist.

## Funding

| Funder | Grant reference number | Author |
|---|---|---|
| Department of Science and Technology, Ministry of Science and Technology | DST-INSPIRE | Santosh Kumar Kuncha |
| Science and Engineering Research Board | J. C. Bose Fellowship | Rajan Sankaranarayanan |
| Department of Biotechnology, Ministry of Science and Technology | Centre of Excellence | Rajan Sankaranarayanan |
| Council of Scientific and Industrial Research | Healthcare Theme project | Rajan Sankaranarayanan |

The funders had no role in study design, data collection and interpretation, or the decision to submit the work for publication.

## Author contributions

Santosh Kumar Kuncha, Conceptualization, Data curation, Formal analysis, Validation, Investigation, Methodology, Writing - original draft, Writing - review and editing; Vinitha Lakshmi Venkadasamy, Gurumoorthy Amudhan, Data curation, Formal analysis, Investigation, Writing - review and editing; Priyanka Dahate, Data curation, Investigation, Writing - review and editing; Sankara Rao Kola, Validation, Investigation, Writing - review and editing, Mass Spec data analysis; Sambhavi Pottabathini, Formal analysis, Investigation, Writing - review and editing; Shobha P Kruparani, P Chandra Shekar, Data curation, Formal analysis, Validation, Methodology, Writing - review and editing; Rajan Sankaranarayanan, Conceptualization, Resources, Data curation, Formal analysis, Supervision, Funding acquisition, Validation, Methodology, Writing - original draft, Project administration, Writing - review and editing

## Author ORCIDs

Santosh Kumar Kuncha (iD) https://orcid.org/0000-0002-2538-8342
Vinitha Lakshmi Venkadasamy (iD) https://orcid.org/0000-0001-7236-2615
Gurumoorthy Amudhan (iD) https://orcid.org/0000-0002-6974-6703
Sambhavi Pottabathini (iD) https://orcid.org/0000-0002-7749-3400
Shobha P Kruparani (iD) http://orcid.org/0000-0002-8955-1647
Rajan Sankaranarayanan (iD) https://orcid.org/0000-0003-4524-9953

## Decision letter and Author response

Decision letter https://doi.org/10.7554/eLife.58118.sa1
Author response https://doi.org/10.7554/eLife.58118.sa2

# Additional files

## Supplementary files

- Transparent reporting form

## Data availability

All the data used in the manuscript is available as source files.

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
