## [Decision Letter]

**Acceptance summary:**

The results are interesting and significant in showing that threonyl-tRNA synthetase can edit misacylated Ala-tRNA^Thr^ in addition to Ser-tRNA^Thr^, and that this activity is inactivated by hydrogen peroxide; and in providing evidence that the parallel editing activity of Animalia-specific-tRNA Deacylase (ATD), which is insensitive to hydrogen peroxide, becomes important in cells treated with hydrogen peroxide for suppressing proteotoxic stress. It is also interesting that the appearance of ATD in evolution of Animalia appears to be coincident with that of tRNAs harboring the G4-U69 base pair that renders them prone to misacylation.

**Decision letter after peer review:**

Thank you for submitting your article "Genomic innovation of ATD alleviates mistranslation associated with multicellularity in Animalia" for consideration by *eLife*. Your article has been reviewed by three peer reviewers, one of whom is a member of our Board of Reviewing Editors, and the evaluation has been overseen by James Manley as the Senior Editor The following individuals involved in review of your submission have agreed to reveal their identity: Michael Ibba (Reviewer #2); Christopher Francklyn (Reviewer #3).

The reviewers have discussed the reviews with one another and the Reviewing Editor has drafted this decision to help you prepare a revised submission.

Summary:

It was known previously that ThrRS can edit misacylated Ser on tRNA^Thr^. This group previously showed that AlaRS can misacylate tRNA^Thr^ for tRNA^Thr^ isoacceptors containing a (G4-U69) pair in the acceptor stem, and that the enzyme ATD can deacylate Ala from such Ala-tRNA^Thr^ molecules. They hypothesized that ATD could serve a proofreading role in the cell. In this report, they show that ThrRS is also capable of deacylating Ala-tRNA^Thr^, apparently the first known case of one synthetase correcting the error made by another; and this activity is widespread among Animalia. They go on to show that this editing activity of ThrRS is inactivated in vitro by H_2_O_2_, whereas ATD's comparable activity is insensitive to H_2_O_2_, leading to the prediction that ATD should be critical for preventing Thr to Ala substitutions in proteins in cells undergoing oxidative stress. They test this by knocking out ATD with CRISPR and finding that this confers reduced cell viability on H_2_O_2_ treatment, which could be complemented by WT but not catalytically inactive ATD expression. They also observe increased Hsp70 expression in the ATD^-/-^ cells on H_2_O_2_ treatment, consistent with increased proteotoxic stress. Supporting this interpretation, they find increased aggregation of a specialized GFP reporter that serves as a reporter for mistranslation; and by mass-spec detected GFP peptides with Thr to Ala changes isolated from ATD^-/-^ mutant cells but not from WT cells. Finally, they present bioinformatics results indicating the appearance of ATD in the Choanoflagellates in parallel with the appearance of tRNAs containing the G-U pair in the acceptor stem that is expected to confer their misacylation with Ala by AlaRS and engender a requirement for ATD editing function. These evolutionary data support a role for ATD in editing Ala misacylation events.

Major revisions:

1) Provide a modified Discussion that addresses (a) the fact that the effect of H_2_O_2_ on cells is not prolonged and so your results might apply only to acute oxidative stress; and also the relationship of H_2_O_2_-induced oxidative stress to the activation of NADPH oxidase; and (b) the rationale for use of Hsp70 induction as a marker for induction of the UPF in oxidative stress, noting the known ties between Hsp70 expression and impairment of the 26S proteosome, and with protein misfolding, particularly in neurodegenerative diseases. (R. Morimoto has reviewed this extensively).

2) Comment on the observation that gamete producing cells (testis and ovaries) are the tissues where ATD is most highly expressed, which might be linked to tissue-specialized responses and sensitivity to oxidative stress.

3) Make the appropriate revisions to respond to the other major points raised by the reviewers.

Reviewer #1:

The results are interesting and significant in showing that ThrRS can edit Ala-tRNA^Thr^ errors in addition to Ser-tRNA^Thr^ errors, and that this activity is inactivated by H_2_O_2_; and in providing evidence that the parallel editing activity of ATD, which is insensitive to H_2_O_2_, becomes important in cells treated with H_2_O_2_ for suppressing proteotoxic stress. It is also interesting that the appearance of ATD in evolution of Animalia appears to be coincident with that of G4-U69 tRNAs.

Reviewer #2:

In the present manuscript, the authors follow up on a previous report from their group (Kuncha et al., 2018.) in which they identified the Animalia-specific tRNA deacylase (ATD). They showed that ATD has enzymatic activity toward mis-aminoacylated Ala-tRNA^Thr^ substrates. Interestingly, the accumulation of this mis-aminoacylated tRNA species does not arise from errors in mis-activation of non-cognate substrates, but rather through mis-selection of the tRNA. Alanyl-tRNA synthetase (AlaRS) utilizes a G3-U70 base pair in the acceptor stem of tRNA^Ala^ for proper recognition. Previous efforts from this group and others have highlighted that in higher eukaryotes, many of the tRNA^Thr^ genes encode a G4-U69 base pair that can be mis-selected by eukaryotic AlaRS. While much of the previous work on ATD focused on the biophysical and biochemical interactions with its substrates, in this report, the authors sought to identify the role of ATD in vivo.

To characterize the function of ATD in vivo, the authors first generated an ATD knockout strain of HEK293T cells. The authors report that the ATD knockout alone caused no discernable phenotype and elicited no induction of the unfolded protein response (UPR) as determined by Western blotting of Hsp70. To ensure this was not due to an artifact in the specific cell type that was selected, the authors show that ATD is expressed across a wide array of tissue types in mice and expressed in various cell lines, which would presumably suggest that it likely is playing an important role in the cell.

As the ATD knockout line displayed no obvious cellular defects, the authors believed that another redundant proofreading factor may also act on Ala-tRNA^Thr^ substrates. One of the most interesting findings from this report was that threonyl-tRNA synthetase (ThrRS) has proofreading activity on Ala-tRNA^Thr^. It has been known for ~20 years that ThrRS utilizes post-transfer proofreading activity to correct mis-aminoacylated Ser-tRNA^Thr^, but the determination that ThrRS also has trans proofreading activity for a different aminoacyl-tRNA substrate was quite unexpected. This result suggests that both ThrRS and ATD can both act on mis-selected Ala-tRNA^Thr^.

It was previously shown that *E. coli* ThrRS is susceptible to oxidation at Cys182. This conserved cysteine is responsible for coordinating the 3' end of the tRNA in the ThrRS proofreading domain, and upon oxidation, loses its deacylase activity for Ser-tRNA^Thr^. The authors then wanted to determine if ThrRS oxidation also perturbed Ala-tRNA^Thr^ deacylation. In fact, oxidative stress does cause an inactivation of ThrRS proofreading on the mis-alanylated tRNA species. This observation suggested that a primary role of ATD may be to deacylate this aminoacyl-tRNA during cellular oxidation.

The ATD knockout strains were revisited and treated with hydrogen peroxide to induce oxidative stress. Unlike the observations in Figure 1, upon oxidation, ATD is required to maintain cell viability and prevent the induction of the UPR. This model suggests that upon oxidation, ThrRS proofreading becomes inactivated thus making ATD the singular factor to deacylate Ala-tRNA^Thr^. To determine if the loss of ATD activity cause protein mistranslation and aggregation, the authors used both a GFP-based reporter and mass spectrometry to show that translational errors were occurring and disrupting proteostasis. An interesting secondary result from these experiments was that the authors could show that the predicted ablation of eukaryotic ThrRS proofreading during oxidative stress led to quantifiable changes in Thr to Ser protein mistranslation in eukaryotic cells. This observation correlates with previous studies in bacteria and will be an important result for those interested in studying mistranslation globally.

Having determined a role for ATD in higher eukaryotes, the authors sought to identify the ancestor for the evolutionary acquisition of ATD. As a model, the authors used the Choanoflagellate *Salpingoecarosetta*. *S. rosetta* is a unicellular organism but encodes for cell cycle and other factors reminiscent of higher eukaryotic physiology. *S. rosetta* ATD had deacylase activity on both Ala-tRNA^Thr^ and Ala-tRNA^Ala^ tRNAs suggesting this protein is more promiscuous for its substrates. This organism does not encode a G4-U69 containing tRNA^Thr^ gene but does however have a tRNA^Gln^ gene which does share this identity element. While not nearly as robust as the previous data, the authors provide some evidence that *S. rosetta* ATD may be important for preventing infrequent Ala-tRNA^Gln^ mis-aminoacylation events.

The aforementioned report is well written and provides new insight into the role of freestanding proofreading factors in higher eukaryotes. The authors utilized a combination of cellular and biochemical approaches to tell a convincing narrative which should allow for future investigation. Aside from a few comments listed below, I would recommend this manuscript for publication in *eLife*.

1) The Western blots in Figure 1C/D and Figure 1—figure supplement 2 are somewhat difficult to interpret. While I agree that they do show the presence of ATD in all of the tissue/cell types, the loading controls are inconsistent. I am less concerned with the technical reproducibility of the sample loading across the different lanes, but more interested in the striking differences in ATD levels across those samples, especially if they were normalized to GAPDH. In those different tissue types, do you believe that GAPDH has varied expression, or do you think the differences in ATD correlate to a more important tissue-specific role of this factor? Furthermore, In Figure 1—figure supplement 2, the figure legend states that protein was normalized by "ATD based normalization" but it is not clear what that means as ATD levels in that blot also vary widely. This technical approach should be reworded for clarity. As a minor comment, the figure legend says that β-actin was used as a loading control, but the figure is labeled as GAPDH.

2) Do the authors have any predicted regulatory mechanism for the elevated ATD levels shown in Figure 5C? Elaboration of this data would be an interesting discussion topic.

3) In Figure 3—figure supplement 4, it is interesting that at position 162, there is variability at that residue with some organisms encoding phenylalanine and others with tyrosine. I am curious if you observed any differences in activity with organisms encoding one or the other amino acid? I am imagining a scenario in which encoding Phe could allow for potential modulation of activity during oxidative stress if that position is accessible and can be modified to tyrosine. Alternatively, oxidized Phe (i.e. tyrosine) may be a more active variant and could have become fixed in those species. It is difficult to interpret any differences in Figure 3—figure supplement 3, but I was wondering if you had observed any other differences between those enzymes? It may speak to an interesting co-regulation of elevated ATD activity during oxidation while ThrRS activity goes down. If you have any data to suggest that may be possible, it would be a very nice discussion point.

Reviewer #3:

The central focus of this manuscript is the identification of conditions under which the Animalia specific tRNA deacylase (ATD) may be necessary for deacylation of tRNAs mis-activated with alanine. The evidence that ThrRS contains deacylation activity is solid, as is the evidence that ATD retains deacylation activity in the presence of H_2_O_2_ in vitro. The weakest part of the manuscript concerns the evidence that ATD and ThrRS functions as a de-acylase under oxidative stress conditions. The evidence that ATD reduces mistranslation is promising, but perhaps still a little preliminary. Overall, this manuscript provides additional characterization of the ATD enzyme, but lacks the decisive experiment that would signal a major advance in our understanding of mechanisms of protein quality control.

Specific comments and concerns:

1) Concern: The lack of quantitation of specific activity of deacylase makes it difficult to assess the extent of the protective effect of ThrRS with regard to clearing Ala-tRNA^Thr^. The argument would be stronger if the authors were able to provide a quantitative comparison of deacylation rates, taking into account enzyme concentrations. This information is likely available from direct fitting of the progress curves that are provided, without performing new experiments.

2) Comment: The retention of Ala-tRNA^Thr^ editing across all ThrRS is not that surprising, when you consider the specific contacts in the ThrRS editing site. While there are interactions that promote Ser-tRNA^Thr^ binding in the editing site, there is no chemical basis to discriminate against Ala in this pocket; hence, this activity is inherent in these enzymes. It also needs to be mentioned that, in prokaryotes, the levels of Ala-tRNA^Thr^ are likely to be significantly lower as a result of the G3:U70 identity element requirement for AlaRS in these taxa. Thus, there Ala-tRNA^Thr^ editing may have little biological impact in the parokaryotic context. The narrative could be modified to include a more nuanced discussion of this issue.

3) Concern: A big concern I have is the use of H_2_O_2_ as a stimulant to induce oxidative stress. It is a good first step but shouldn't be the whole story. The problem is that H_2_O_2_ has a relatively short life when administered to cells, and the short-term physiological effects that it induces might not be reflective of longer-term adaptation. (inducers of the NADPH oxidase should be considered.) The data suggesting that ATD is essential during oxidative stress would have been more convincing if it were supported by independent measurements of oxidative stress in the cells. The β-mercaptoethanol experiments provide indirect support of the hypothesis at best. A difficulty here is that it is not easy to formally block deacylation activity of ThrRS without blocking aminoacylation. A Potential method to be considered involves use of redox sensitive dye molecules, which can provide quantitative information. The authors might also consider looking at conditions under which stress granules form, given the interest in mistranslation. This represents the only concern where additional experimental work might be warranted.

4) Comment: While I would have liked to see a SILAC experiment formally comparing the extents of mistranslation ATD+ and ATD- cells, the mass spec data shown are promising and support the hypothesis that ATD forms this important editing function.

5) Comment: I really like the bioinformatics section of the paper. This was a very thoughtful addition to the study, with nice broader biological implications. Going forward, it will be very interesting to investigate the evolutionary emergence of other components in the protein synthesis quality control apparatus. The approach here is readily adaptable to the analysis of other translation genes.

---

## [Author Response]

Major revisions:1) Provide a modified Discussion that addresses (a) the fact that the effect of H_2_O_2_ on cells is not prolonged and so your results might apply only to acute oxidative stress; and also, the relationship of H_2_O_2_-induced oxidative stress to the activation of NADPH oxidase;

We agree with the reviewers that H_2_O_2_ induces acute oxidative stress, while NADPH oxidase is known to induce chronic oxidative stress. The toxicity experiments were done to bring out the role of ATD in the presence of oxidative stress. However, in the context of ATD, more targeted/focused studies to find out the subtle oxidative stress response using different NADPH oxidase activating molecules across different cell lines will be the focus of future studies. This is now included in the revised manuscript as: “The current work involves the use of acute oxidative stress (H_2_O_2_) and further prompts to look at the behaviour of ATD knockout cells in the presence of chronic oxidative stress such as activators of NADPH oxidase.”

b) the rationale for use of Hsp70 induction as a marker for induction of the UPF in oxidative stress, noting the known ties between Hsp70 expression and impairment of the 26S proteosome, and with protein misfolding, particularly in neurodegenerative diseases. (R. Morimoto has reviewed this extensively).

We thank the reviewers for this suggestion. We have now included this in the revised manuscript as: “Hsp70 is a chaperone which is upregulated in response to protein misfolding and hence used as a marker to study proteostasis stress and is also implicated in aging and neurodegenerative diseases (Gupta et al., 2011; Hartl, 1996; Mosser et al., 2000; Rampelt et al., 2012; Sala et al., 2017).”

2) Comment on the observation that gamete producing cells (testis and ovaries) are the tissues where ATD is most highly expressed, which might be linked to tissue-specialized responses and sensitivity to oxidative stress.

We have also noted the higher expression of ATD in reproductive organs and it is very likely due to the high oxidative stress in these tissues. This could be linked to the regulation of ATD levels under different conditions that need to be probed further. This aspect has been now included in the revised manuscript: “ATD expression levels are very likely modulated by the level of oxidative stress as seen in the case of reproductive tissues (Figure 1C, D), which also suggests a mode of regulation via feedback mechanism.”

3) Make the appropriate revisions to respond to the other major points raised by the reviewers.

We have now included all the revisions in the revised manuscript as mentioned below.

Reviewer #2:[…]1) The Western blots in Figures 1C/D and Figure 1—figure supplement 2 are somewhat difficult to interpret. While I agree that they do show the presence of ATD in all of the tissue/cell types, the loading controls are inconsistent. I am less concerned with the technical reproducibility of the sample loading across the different lanes, but more interested in the striking differences in ATD levels across those samples, especially if they were normalized to GAPDH. In those different tissue types, do you believe that GAPDH has varied expression, or do you think the differences in ATD correlate to a more important tissue-specific role of this factor? Furthermore, In Figure 1—figure supplement 2 , the figure legend states that protein was normalized by "ATD based normalization" but it is not clear what that means as ATD levels in that blot also vary widely. This technical approach should be reworded for clarity. As a minor comment, the figure legend says that β-actin was used as a loading control, but the figure is labeled as GAPDH.

Please also see the response to major revision 2, above. The house-keeping protein GAPDH is the most widely used loading control and is expected not to vary much among different tissues. Therefore, we agree with the reviewer that ATD expression is high in testis and ovaries hinting at a tissue-specific role, which is known to have high levels of oxidative stress, and the same is now included in the revised manuscript: “ATD expression levels are very likely modulated by the level of oxidative stress as seen in the case of reproductive tissues (Figure 1C, D), ….”

By stating ATD based normalization, we intended to say that due to high expression of ATD in a few cells/tissues we had to increase the overall protein load of samples in which ATD expression is low, this approach has allowed us to detect ATD in tissues expressing at very low levels. However, the usage of the word ATD-based normalization is not appropriate and hence this is now removed from the manuscript. We have also changed the figure legend of Figure 1—figure supplement 2 from β-actin to GAPDH and reworded as “…ATD protein was used as a positive control and GAPDH as a loading control.” We are very thankful to the reviewer for these suggestions.

2) Do the authors have any predicted regulatory mechanism for the elevated ATD levels shown in Figure 5C? Elaboration of this data would be an interesting discussion topic.

We acknowledge the reviewer for this pertinent suggestion. The increased expression of ATD upon H_2_O_2_ treatment, suggest a possible oxidative stress-induced positive feedback loop for the induction of ATD. This is now included in the discussion of the revised manuscript: “…which also suggests a mode of regulation via feedback mechanism.”

3) In Figure 3—figure supplement 4, it is interesting that at position 162, there is variability at that residue with some organisms encoding phenylalanine and others with tyrosine. I am curious if you observed any differences in activity with organisms encoding one or the other amino acid? I am imagining a scenario in which encoding Phe could allow for potential modulation of activity during oxidative stress if that position is accessible and can be modified to tyrosine. Alternatively, oxidized Phe (i.e. tyrosine) may be a more active variant and could have become fixed in those species. It is difficult to interpret any differences in Figure 3—figure supplement 3, but I was wondering if you had observed any other differences between those enzymes? It may speak to an interesting co-regulation of elevated ATD activity during oxidation while ThrRS activity goes down. If you have any data to suggest that may be possible, it would be a very nice discussion point.

As per the reviewer’s suggestion, we have now analysed our data in the light of Phe oxidation as explained below. In the current study, we have biochemical data of ATD from multiple organisms (*H. sapiens*, *M. musculus*, *G. gallus*, *D. rerio*, and *H. vulgaris*). Of the five ATDs tested, Human ATD (HsATD) has a Phe, while all others harbour Tyr at this position (162 in MmATD). The biochemical data (Figure 3—figure supplement 3) does not show any notable difference in activity between the Y162 and F162 containing variants of ATD with and without H_2_O_2_. However, such a kind of regulation is very unlikely due to the lack of Phe invariance at this position (Author response image 1).

**Author response image 1. sa2fig1:** Conservation of Y162 of ATD. The image depicts the frequency plot of TNGPY/FTH motif generated using 200 sequences of ATD.

Reviewer #3:[…]Specific comments and concerns:1) Concern: The lack of quantitation of specific activity of deacylase makes it difficult to assess the extent of the protective effect of ThrRS with regard to clearing Ala-tRNA^Thr^. The argument would be stronger if the authors were able to provide a quantitative comparison of deacylation rates, taking into account enzyme concentrations. This information is likely available from direct fitting of the progress curves that are provided, without performing new experiments.

As correctly suggested by the reviewer, we have now calculated the rates of deacylation, and included them as a table in the revised manuscript.

2) Comment: the retention of Ala-tRNA^Thr^ editing across all ThrRS is not that surprising, when you consider the specific contacts in the ThrRS editing site. While there are interactions that promote Ser-tRNA^Thr^ binding in the editing site, there is no chemical basis to discriminate against Ala in this pocket; hence, this activity is inherent in these enzymes. It also needs to be mentioned that, in prokaryotes, the levels of Ala-tRNA^Thr^ are likely to be significantly lower as a result of the G3:U70 identity element requirement for AlaRS in these taxa. Thus, there Ala-tRNA^Thr^ editing may have little biological impact in the parokaryotic context. The narrative could be modified to include a more nuanced discussion of this issue.

As correctly pointed out by the reviewer, the bacterial AlaRS is not ambiguous in tRNA selection and therefore charges only tRNAs harbouring G3-U70, but not G4-U69. Since G3-U70 is very specific to tRNA^Ala^, the problem of tRNA mis-selection does not exist in prokaryotes and therefore ATD is absent in these organisms. This is now more elaborately mentioned in the revised manuscript: “…even though neither Bacteria (bacterial AlaRS is discriminatory and charges only tRNAs containing G3-U70 (Sun et al., 2016)) nor lower eukaryotes (which lack tRNAs containing G4-U69) possess the problem of tRNA mis-selection.”

3) Concern: A big concern I have is the use of H_2_O_2_ as a stimulant to induce oxidative stress. It is a good first step but shouldn't be the whole story. The problem is that H_2_O_2_ has a relatively short life when administered to cells, and the short-term physiological effects that it induces might not be reflective of longer-term adaptation. (inducers of the NADPH oxidase should be considered.) The data suggesting that ATD is essential during oxidative stress would have been more convincing if it were supported by independent measurements of oxidative stress in the cells. The β-mercaptoethanol experiments provide indirect support of the hypothesis at best. A difficulty here is that it is not easy to formally block deacylation activity of ThrRS without blocking aminoacylation. A Potential method to be considered involves use of redox sensitive dye molecules, which can provide quantitative information. The authors might also consider looking at conditions under which stress granules form, given the interest in mistranslation. This represents the only concern where additional experimental work might be warranted.

We thank the reviewer for bringing up this point. As a first major study, we have used H_2_O_2_ to bring out the role of ATD in the presence of acute oxidative stress as has been followed generally by others in the field. As correctly stated by the reviewer, the H_2_O_2_ toxicity experiments are further supported by the toxicity caused due to depletion of reducing agent (β-mercaptoethanol) in mouse ES cells. However, intricate studies would be required in the future to look at the subtle effect of oxidative stress on ATD knockout cells in the presence of NADH oxidase activators in combination with assessing the cellular ROS (using redox dyes). We thank the reviewer for the suggestion of using FlucDM, the sensor of proteome stress, as a screen for testing different conditions in which ATD plays an essential role in maintaining proteome homeostasis, which we intend to use in our future studies. Based on the reviewer’s comments we have now included these aspects in our revised manuscript.

4) Comment: While I would have liked to see a SILAC experiment formally comparing the extents of mistranslation ATD+ and ATD- cells, the mass spec data shown are promising and support the hypothesis that ATD forms this important editing function.

We agree with the reviewer and this will be our future course of study in different cell lines using different ROS inducers to precisely quantify the extent of mistranslation.

5) Comment: I really like the bioinformatics section of the paper. This was a very thoughtful addition to the study, with nice broader biological implications. Going forward, it will be very interesting to investigate the evolutionary emergence of other components in the protein synthesis quality control apparatus. The approach here is readily adaptable to the analysis of other translation genes.

We thank the reviewer for this comment.